# SHAPO: Sharpness-Aware Policy Optimization for Safe Exploration

**Kaustubh Mani** [*]
Université de Montréal
Mila - Québec AI Institute

**Yann Pequignot**
Université Laval
IID - Institut Intelligence et Données

**Vincent Mai**
LawZero

**Liam Paull**
Université de Montréal
Mila - Québec AI Institute
CIFAR AI Chair

## Abstract

Safe exploration is a prerequisite for deploying reinforcement learning (RL) agents in safety-critical domains. In this paper, we approach safe exploration through the lens of epistemic uncertainty, where the actor's sensitivity to parameter perturbations serves as a practical proxy for regions of high uncertainty. We propose *Sharpness-Aware Policy Optimization* (*SHAPO*), a sharpness-aware policy update rule that evaluates gradients at perturbed parameters, making policy updates pessimistic with respect to the actor's epistemic uncertainty. Analytically we show that this adjustment implicitly reweighs policy gradients, amplifying the influence of rare unsafe actions while tempering contributions from already safe ones, thereby biasing learning toward conservative behavior in under-explored regions. Across several continuous-control tasks, our method consistently improves both safety and task performance over existing baselines, significantly expanding their Pareto frontiers.

## 1 Introduction

Deploying reinforcement learning (RL) agents in safety-critical environments poses unique challenges due to the need for *safe exploration*: the process by which an agent collects informative experience while avoiding catastrophic failures (Bharadhwaj et al., 2020; Achiam et al., 2017; Sootla et al., 2022). A fundamental difficulty arises during early training, when the agent inevitably visits states about which it has limited knowledge, i.e., regions of *high epistemic uncertainty*. In these regions, the predictions of learned policies and value functions are unreliable, and exploratory actions may cause unsafe or irreversible outcomes.

In practice, safe exploration depends on the policies an algorithm visits while learning the task, so the choice of update rule matters: each gradient step should hedge against unreliable estimates in parts of the state–action space visited under the current policy. This connects safe exploration to the agent's epistemic uncertainty—uncertainty about model parameters that is largest where on-policy data are scarce—suggesting that policy updates themselves should be pessimistic with respect to that epistemic uncertainty.

Prior work has largely handled uncertainty on the critic side (ensembles (Lakshminarayanan et al., 2017; Osband et al., 2016; Bykovets et al., 2022), dropout (Gal & Ghahramani, 2016), or distributional methods (Bellemare et al., 2017; Luis et al., 2024) combined with risk measures), which helps quantify variability in returns but does not expose a practical, optimization-ready notion of actor's epistemic uncertainty. Maintaining an explicit posterior over policy parameters is generally infeasible for deep on-policy RL, making actor-side risk aversion difficult in practice. As a result, while critic-side approaches are common, we observe a gap in methods for policy updates that carefully account for the actor's parameter uncertainty.

---

[*]Corresponding author: kaustubh3095@gmail.com

We address this gap by introducing a novel approach to policy update guided by gradients evaluated at an adjusted parameter: *Sharpness Aware Policy Optimization* (*SHAPO*). Our method builds upon sharpness-aware optimization (Foret et al., 2020; Kim et al., 2022b), which incorporates the sharpness of the loss landscape to enhance generalization. We reinterpret this adjustment through the perspective of epistemic uncertainty, demonstrating that it induces a systematically pessimistic bias with respect to the actor's performance. In a simplified analytical setting, we show that *SHAPO* modifies the treatment of low-probability actions: those deemed unsafe are assigned greater weight in the policy update, whereas those identified as safe are downweighted. Empirically, *SHAPO* expands the safety–efficiency Pareto frontier across a range of on-policy Safe RL baselines in the Safety-Gym (Ray et al., 2019) and MuJoCo (Todorov et al., 2012) environments, while simultaneously reducing cumulative failures and mitigating heavy-tailed episodic cost distributions. Ablation studies further indicate that perturbations defined in the Fisher metric are superior to their Euclidean counterparts, and that applying sharpness-aware optimization to the actor is more consequential for safe exploration than its application to the critic.

## 2 BACKGROUND

### 2.1 CONSTRAINED MARKOV DECISION PROCESSES

A *Constrained Markov Decision Process* (CMDP) (Altman, 2021) is a tuple

$$\mathcal{M} = (\mathcal{S}, \mathcal{A}, T, r, c, \gamma, \mu_0),$$

with state space $\mathcal{S}$, action space $\mathcal{A}$, transition kernel $T$, reward function $r : \mathcal{S} \times \mathcal{A} \to \mathbb{R}$, nonnegative cost function $c : \mathcal{S} \times \mathcal{A} \to \mathbb{R}_+$, discount factor $\gamma \in [0, 1)$, and initial state distribution $\mu_0$. A stationary policy $\pi$ induces discounted returns

$$J_r(\pi) = \mathbb{E}_\pi \Big[ \sum_{t=0}^\infty \gamma^t r(s_t, a_t) \Big], \quad J_c(\pi) = \mathbb{E}_\pi \Big[ \sum_{t=0}^\infty \gamma^t c(s_t, a_t) \Big],$$

and the objective is

$$\pi^* = \underset{\pi \in \Pi_C}{\mathrm{argmax}}\, J_r(\pi), \Pi_\mathcal{C} = \big\{ \pi \in \Pi : J_c(\pi) \le \beta \big\}. \tag{1}$$

for a constraint threshold $\beta \ge 0$.

A common approach consists in introducing a Lagrangian objective,

$$J_\lambda(\pi_\theta) = J_r(\pi_\theta) - \lambda\big(J_c(\pi_\theta) - \beta\big), \tag{2}$$

with multiplier $\lambda \ge 0$ adapted to enforce the constraint.

We write $Q_\lambda^\pi(s, a) = E_{\tau \sim \pi}[R_\lambda(\tau)|s_0 = s, a_0 = a]$ for the action-value function associated with $\pi$, where $R_\lambda(\tau) = \sum_{t=0}^\infty \gamma^t \big(r(s_t, a_t) - \lambda c(s_t, a_t)\big)$. Similarly, we write $V_\lambda^\pi(s) = E_{a \sim \pi(\cdot|s)}[Q_\lambda^\pi(s, a)]$ for the value function. Finally, for a policy $\pi$, we write $d^\pi(s)$ for the discounted state distribution given by $d^\pi(s) = (1 - \gamma) \sum_{t=0}^\infty \gamma^t Pr(s = s_t|\pi)$.

### 2.2 POLICY OPTIMIZATION

A standard way to learn policies that maximize expected return consists of iteratively improving a policy until convergence to optimum. We provide here some notations and results from Kakade & Langford (2002); Schulman et al. (2015). Given a current policy $\pi_0$ we seek to find a policy $\pi$ that improves on $\pi_0$. The improvement of a policy $\pi$ over $\pi_0$ can be expressed as:

$$J_\lambda(\pi) - J_\lambda(\pi_0) = \mathbb{E}_{s \sim d^\pi} \mathbb{E}_{a \sim \pi(a|s)} A_\lambda^0(s, a)$$

where $A_\lambda^0(s, a) = Q_\lambda^{\pi_0}(s, a) - V_\lambda^{\pi_0}(s)$ quantifies the advantage of taking action $a$ compared to the average action under policy $\pi_0$.

This improvement is an expectation over the state distribution of the unknown policy $\pi$, so, in practice, it is replaced by the the following quantity that can be estimated using only experience collected with the current $\pi_0$:

$$L_{\pi_0}^\lambda(\pi) = \mathbb{E}_{s \sim d^{\pi_0}} \mathbb{E}_{a \sim \pi_0(a|s)} \frac{A_\lambda^0(s, a)}{\pi_0(a|s)} \pi(a|s). \tag{3}$$

However in order for a positive $L_{\pi_0}^\lambda(\pi)$ to effectively represent an improvement, we need to ensure that $\pi$ does not differ too much from $\pi_0$ . As a consequence, to generate an updated policy $\pi$, the natural gradient is found as the solution to the following optimization problem:

$$
\begin{aligned}
&\underset{\pi}{\text{maximize}}\ L_{\pi_0}^\lambda(\pi) \\
&\text{subject to}\ D_{\mathrm{KL}}^{\max}(\pi_0, \pi) \leq \delta
\end{aligned}
\tag{4}
$$

where $D_{\mathrm{KL}}^{\max}(\pi_0, \pi) = \max_s \mathrm{KL}(\pi_0(\cdot|s)\|\pi(\cdot|s))$ is the maximum of Kullback-Leibler divergence over states and $\delta > 0$ specifies the trust region constraint.

We recall the following result adapted from (Schulman et al., 2015, Theorem 1) that bounds the improvement (or deterioration) of a policy $\pi$ over the current policy $\pi_0$ in terms of $L_{\pi_0}^\lambda(\pi)$:

**Theorem 1.** *For $\alpha = D_{KL}^{max}(\pi_0, \pi)$, we have*

$$
L_{\pi_0}^\lambda(\pi) - B_\alpha \leq J_\lambda(\pi) - J_\lambda(\pi_0) \leq L_{\pi_0}^\lambda(\pi) + B_\alpha \quad \text{with} \quad B_\alpha = -\alpha \frac{4\gamma}{(1-\gamma)^2} \max_{s,a} |A_\lambda^0(s,a)|.
$$

Intuitively, this result indicates that the further $\pi$ is from $\pi_0$ (in the sense of $D_{\mathrm{KL}}^{\max}$), the more $L_{\pi_0}^\lambda(\pi)$ must deviate from $0 = L_{\pi_0}^\lambda(\pi_0)$[1] to ensure a true improvement (or deterioration).

We now consider a parametrized policy $\pi_\theta$, let $\pi_0 = \pi_{\theta_0}$ and overload our previous notation by using $\theta$ instead of $\pi_\theta$ when there is no risk of confusion. Since the maximum in $D_{\mathrm{KL}}^{\max}(\pi_0, \pi_\theta)$ is intractable, the expectation $\overline{D}_{\mathrm{KL}}^{\pi_0}(\pi_0, \pi_\theta) = \mathbb{E}_{s \sim d^{\pi_0}} \mathrm{KL}(\pi_0(\cdot|s)\|\pi_\theta(\cdot|s))$ is used in practice. The second-order approximation of this function of $\theta$ around $\theta_0$ is given by $\overline{D}_{\mathrm{KL}}^{\pi_0}(\pi_0\|\pi_{\theta_0+\epsilon}) \approx \frac{1}{2}\epsilon^T \mathbf{F}_{\theta_0} \epsilon$ where $\mathbf{F}_{\theta_0}$ denotes the Fisher Information Matrix defined by $\mathbf{F}_{\theta_0} = \mathbb{E}_{s \sim d^{\pi_0}} \mathbb{E}_{a \sim \pi_0(a|s)} (\nabla_\theta \log \pi_\theta(a|s)|_{\theta=\theta_0})(\nabla_\theta \log \pi_\theta(a|s)|_{\theta=\theta_0})^T$ (Kullback, 1997, p. 26). Now, using a linear approximation $L_{\pi_0}^\lambda(\theta) = g^T\theta$ of the loss function around $\theta_0$ leads to the following optimization problem:

$$
U_{\mathbf{F}_{\theta_0}}(g, \delta) = \underset{\frac{1}{2}\epsilon^T \mathbf{F}_{\theta_0} \epsilon \leq \delta}{\arg\max} \langle g, \epsilon \rangle.
\tag{5}
$$

TRPO (Schulman et al., 2015) uses $g = \nabla_\theta L_{\pi_0}^\lambda(\theta)|_{\theta=\theta_0}$ for the parameter update. We observe that relying on the gradient $g$ of the loss $L_{\pi_0}^\lambda$ may result in an overconfident update, as it does not take into account the epistemic uncertainty associated with $\pi_0$.

## 2.3 SHARPNESS-AWARE OPTIMIZATION

Sharpness-Aware Minimization (SAM) (Foret et al., 2020) is a general framework that aims to bias solutions towards flat regions of the loss or utility landscape. In general terms, for a utility function $\mathcal{L}(\pi)$ that we seek to maximize for $\pi \in \Pi$, SAM framework proposes the following modified optimization problem:

$$
\max_{\pi \in \Pi} \min_{\tilde{\pi} \in N(\pi)} \mathcal{L}(\tilde{\pi}),
$$

where $N(\pi)$ is some choice of neighborhood of $\pi$. Given a parametrization $\pi_\theta$ with $\theta \in \Theta$, Euclidean neighborhoods in parameter space represent a simple choice that was studied by Foret et al. (2020) and has also applied to RL (Lee & Yoon, 2025). Other neighborhoods based on functional similarity have also been proposed in the supervised learning setting. Fisher-SAM (Kim et al., 2022b) proposes to define the neighborhood of a predictive distribution $\pi : x \mapsto \pi(\cdot|x)$ in terms of expected Kullback-Leibler divergence $N_\delta^{\mathrm{KL}}(\pi) = \{\tilde{\pi} : \mathbb{E}_x[\mathrm{KL}(\pi(\cdot|x)\|\tilde{\pi}(\cdot|x))] \leq \delta\}$ for some $\delta \geq 0$. In practice, maximization of this worst-case value in a neighborhood is achieved by successive ascent on parameter $\theta$ following the gradient of the loss at $\mathcal{L}(\theta + \epsilon)$ where $\epsilon$ is such that $\pi_{\theta+\epsilon} \approx \arg\min_{\tilde{\pi} \in N(\pi)} \mathcal{L}(\tilde{\pi})$ (see Alg. 2 in appendix). In the next section, we adapt the sharpness-aware optimization approach for policy optimization. Furthermore, we show that this method is equivalent to optimizing a risk-aware objective in light of the epistemic uncertainty surrounding the current policy.

---

[1]Note that $L_{\pi_0}^\lambda(\pi_0) = \mathbb{E}_{s \sim d^{\pi_0}} \mathbb{E}_{a \sim \pi_0(\cdot|s)} A_\lambda^0(s,a) = 0$ since $A_\lambda^0(s,a) = Q_\lambda^0(s,a) - \mathbb{E}_{a \sim \pi_0(\cdot|s)}[Q_\lambda^0(s,a)]$

## 3 SAFE EXPLORATION

On-policy deep RL algorithms for CMDPs begin with a random policy $\pi_0$ and iteratively update to $\pi_1, \pi_2, \ldots, \pi_{\text{last}}$. The objective is to maximize $J_r(\pi_{\text{last}})$ subject to $J_c(\pi_{\text{last}}) \leq \beta$. However, during this process the agent incurs a cumulative training cost proportional to $\sum_{i=1}^{\text{last}} J_c(\pi_i)$. The *primary objective* of safe exploration is therefore to minimize this cumulative cost while still learning a policy that yields high expected reward.

A complementary objective is to avoid policies $\pi_i$ that, even if low-cost in expectation, occasionally produce trajectories with very high cost. Writing the discounted trajectory cost as $J_c(\tau) = \sum_{t=0}^{\infty} \gamma^t c(s_t, a_t)$, the goal is to control the tail of the distribution of $J_c(\tau)$, thereby reducing the frequency of rare but catastrophic failures.

We observe that these two objectives extend beyond merely finding a solution to the CMDP. They emphasize not only the importance of maximizing expected rewards while adhering to cost constraints but also the critical need for the policies explored during training to avoid rare high-cost outcomes. In practice, selecting different values of $\lambda$ in the equation 2 results in a trade-off between expected return and total cost. The focus of this work is on safe exploration, which involves advancing this Pareto front.

Policy updates are crucial to the effectiveness of these methods. Given the inherent uncertainties associated with these updates, adopting a pessimistic approach—especially in the context of the agent's epistemic uncertainty—emerges as a logical strategy for achieving safe exploration, which we implement in this work. Specifically, we believe that a safe exploration framework must effectively address rare and potentially unsafe events during the policy update process. By prioritizing this consideration, we aim to enhance both the reliability and safety of the learning process and its outcomes.

## 4 SHARPNESS AWARE POLICY OPTIMIZATION (*SHAPO*)

In the context of policy optimization, SAM framework (Foret et al., 2020) suggests to look for a policy $\pi$ that maximizes

$$\min_{\tilde{\pi} \in N(\pi)} J_\lambda(\tilde{\pi}) \tag{6}$$

instead of just $J_\lambda(\pi)$. This shift in objectives can be intuitively justified at high level as follows. We may prefer a policy $\pi_1$ over $\pi_2$ even if $J_\lambda(\pi_1) < J_\lambda(\pi_2)$ if $\pi_1$ achieves a higher value for the worst-case objective 6. This preference may arise from the understanding that the expected return of policies within the neighborhood $N(\pi_1)$ can provide a more accurate estimate of the actual performance of $\pi_1$. This is particularly relevant because our learning problem may not fully encapsulate real-world complexities, such as environmental dynamics or the real-world implementation of our policy. By focusing on the worst-case expected return of nearby policies, we aim to better align our estimates with the true performance outcomes in realistic settings.

### 4.1 THE POLICY UPDATE

As outlined in the background section, our goal is to learn a policy $\pi$ that maximizes $J_\lambda(\pi)$ by gradually updating the current policy $\pi_{\theta_0}$. This is achieved by solving the optimization problem 5 under a trust region constraint $\delta_{\text{Up}}$ and using the gradient $g = \nabla_\theta L_{\pi_0}^\lambda(\theta)\big|_{\theta=\theta_0}$. We present here our proposed approach, which differs primarily in that it substitutes the direction $g$ with the gradient $\tilde{g}$ of $L_{\pi_0}^\lambda$ computed at $\theta_0 + \epsilon_{\text{Down}}$ where $\epsilon_{\text{Down}}$ is a carefully chosen adjustment.

Given the current policy $\pi_0$, we now outline our proposed method for updating it within the framework of SAM. We have at our disposal the utility function $L_{\pi_0}^\lambda(\theta)$ which estimates the relative improvement (or deterioration) of nearby policies $\pi_\theta$. Our first step therefore consists of solving the minimization problem

$$\min_{\tilde{\pi} \in N(\pi_0)} L_{\pi_0}^\lambda(\tilde{\pi}). \tag{7}$$

We note that choosing an isotropic Euclidean neighborhood in parameter space for $N(\pi_0)$ presents certain challenges. A small distance in parameter space ($\|\theta - \theta_0\|_2$ small) does not necessarily

---

**Algorithm 1** *SHAPO*$(\theta, L, F, \delta_{\text{Down}})$

---

**Require:** parameter $\theta$, objective function $L$, Fisher matrix $F$, constraint $\delta_{\text{Down}}$
  1: Compute the gradient at $\theta$: $g \leftarrow \nabla_\theta L(\theta)|_\theta$
  2: Compute the perturbation: $\epsilon_{\text{Down}} = U_F(-g, \delta_{\text{Down}})$
  3: Compute the gradient at perturbed parameter: $\tilde{g} \leftarrow \nabla_\theta L(\theta)|_{\theta + \epsilon_{\text{Down}}}$
  4: Return *SHAPO* gradient $\tilde{g}$

---

guarantee that $\pi_\theta$ is similar to $\pi_0$ in terms of $D_{\text{KL}}^{\max}$. Consequently, when $L_{\pi_0}^\lambda(\theta) < 0$, it does not ensure that $J_\lambda(\pi_\theta) < J_\lambda(\pi_0)$. In contrast, while selecting a neighborhood of distributionally similar policies better reflects the underlying geometry of the statistical manifold of the parameters (Kim et al., 2022b), it also contributes to ensure the validity of utility function $L_{\pi_0}^\lambda$ in virtue of Theorem 1.

Consequently, to identify a policy $\tilde{\pi}$ that is distributionally close to $\pi_0$ but performs worse than $\pi_0$, we propose to formulate the inner minimization 7 in a manner analogous to the optimization problem 4, namely as

$$\begin{aligned} &\underset{\tilde{\pi}}{\text{minimize}} \; L_{\pi_0}^\lambda(\tilde{\pi}) \\ &\text{subject to } D_{\text{KL}}^{\max}(\pi_0, \tilde{\pi}) \leq \delta_{\text{Down}} \end{aligned} \tag{8}$$

for some trust region constraint $\delta_{\text{Down}} \geq 0$. We therefore estimate the solution to the inner minimization with the policy $\tilde{\pi} = \pi_{\theta_0 + \epsilon_{\text{Down}}}$ where the perturbation $\epsilon_{\text{Down}} = U_{\mathbf{F}_{\theta_0}}(-\nabla_\theta L_{\pi_0}^\lambda(\theta)|_{\theta=\theta_0}, \delta_{\text{Down}})$ is the solution to the optimization problem 5 where $\mathbf{F}_{\theta_0}$ is the Fisher information matrix of our policy parametrization evaluated at $\theta_0$.

Next we proceed to update $\pi_0$ based on the direction provided by the gradient of $L_{\pi_0}^\lambda$ at our found perturbed parameter $\theta_0 + \epsilon_{\text{Down}}$. This step involves solving the maximization problem 5, where the gradient $\nabla_\theta L_{\pi_0}^\lambda(\theta)$ is now evaluated at $\theta_0 + \epsilon_{\text{Down}}$. This results in a parameter update given by $\epsilon_{\text{Up}} = U_{\mathbf{F}_{\theta_0}}(\nabla_\theta L_{\pi_0}^\lambda(\theta)|_{\theta=\theta_0 + \epsilon_{\text{Down}}}, \delta_{\text{Up}})$ where $\delta_{\text{Up}} \geq 0$ is the trust region constraint. The pseudo-code for computing the *SHAPO* gradients is given in Algorithm 1.

### 4.2 Reinterpreting Fisher SAM as Pessimism in Face of Epistemic Uncertainty

In effect, our proposed approach differs from classical local optimization techniques in that the update of the current parameter $\theta_0$ is guided by the gradient evaluated at $\theta_0 + \epsilon_{\text{Down}}$ instead of $\theta_0$. We now provide a different perspective on the perturbation $\epsilon_{\text{Down}}$, arising from a pessimistic viewpoint in the context of epistemic uncertainty.

We propose to model the uncertainty about the parameter $\theta_0$, which defines the current policy $\pi_0 = \pi_{\theta_0}$ using a multivariate normal distribution $Q(\theta) = \mathcal{N}(\theta_0, \Sigma)$ with mean $\theta_0$ and covariance matrix $\Sigma = \frac{1}{n}\mathbf{F}^{-1}$ where $\mathbf{F}^{-1}$ denotes the inverse of the Fisher Information Matrix $\mathbf{F} = \mathbf{F}_{\theta_0}$ evaluated at $\theta_0$. The choice to model epistemic uncertainty this way is motivated by the asymptotic properties established by the Bernstein–von Mises Theorem (van der Vaart, 1998) where $n$ represents the number of independent samples used to estimate the parameters of the model. However, it should be noted that in the case of RL this not simple to compute due to the correlation of samples, as we will elaborate on in Section 4.4. This approach captures the intuitive notion that as we gather more data, our uncertainty about the parameter decreases, and the distribution of our estimates becomes more concentrated around the true value. This choice not only aligns with theoretical foundations but also offers a practical starting point for exploring the implications of epistemic uncertainty in the context of policy optimization.

We observe that this uncertainty about the current parameter $\theta_0$ translates into an uncertainty about the expected return $J_\lambda(\pi_\theta)$ of our current policy, which we can estimate as

$$J_\lambda(\pi_\theta) \approx J_\lambda(\pi_{\theta_0}) + L_{\pi_0}^\lambda(\theta)$$

when $\pi_\theta$ is close to $\pi_0$ in distribution. Writing $g = \nabla_\theta L_{\pi_\theta}^\lambda(\theta)|_{\theta=\theta_0}$ and assuming a first-order approximation $L_{\pi_0}^\lambda(\theta + \epsilon) = \epsilon^T g$ around $\theta_0$, the deviation of expected return is given by $Y = L_{\pi_0}^\lambda(\theta), \theta \sim \mathcal{N}(\theta_0, \frac{1}{n}\mathbf{F}^{-1})$ that follows a normal distribution $\mathcal{N}(0, \sigma^2)$ with variance $\sigma^2 = \frac{1}{n}g^T\mathbf{F}^{-1}g$.

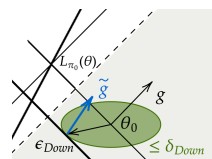 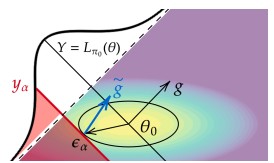 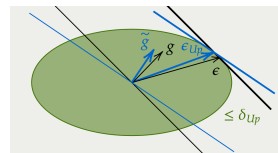

Figure 1: *SHAPO* directs the policy update using the gradient $\tilde{g}$ at an adjusted parameter (Right). Two perspectives on this adjustment are illustrated: (Left) The adjustment aims to minimize the expected return while staying within the trust region defined by $\delta_{\text{Down}}$. (Middle) Parameter uncertainty translates to uncertainty in the expected return, resulting in a maximum likelihood adjustment that remains below the $\alpha$-quantile $y_\alpha$.

While $L^\lambda_{\pi_0}(\theta_0) = 0$ represents the mean of this distribution, parameters for which $L^\lambda_{\pi_0}(\theta)$ falls at, for instance, the 5th percentile of the distribution $Y$ provide a more nuanced perspective on the current expected return. This highlights that, although the mean may indicate a certain level of performance, there are credible scenarios in which performance could be significantly lower. Therefore, adopting a pessimistic standpoint, it is justified to prioritize improving a more conservative expected return when updating $\theta_0$. We demonstrate that evaluating the gradient at $\theta_0 + \epsilon_{\text{Down}}$ to inform the update of $\theta_0$ can be interpreted as optimizing an estimate of the expected return that is pessimistic in face of the parameter uncertainty.

For $\alpha \in (0, \frac{1}{2})$, let $y_\alpha$ denote the $\alpha$-quantile of the random variable $Y$, and $z_\alpha$ denote the $\alpha$-quantile of the normal distribution $\mathcal{N}(0, 1)$. Note that under our first-order approximation of $L^\lambda_{\pi_0}$ the condition $L^\lambda_{\pi_0}(\theta_0 + \epsilon) \leq y_\alpha$ corresponds to $\epsilon^T g \leq y_\alpha$. The following proposition links this epistemic uncertainty perspective with the parameter perturbation described in the previous section. It is illustrated in Fig. 1.

**Proposition 2.** *Let $Q(\theta) = \mathcal{N}(\theta_0, \frac{1}{n}\mathbf{F}^{-1})$. For every $\alpha \in (0, \frac{1}{2})$, the solution $\epsilon_\alpha$ to the optimization problem*

$$\underset{\epsilon}{\text{maximize}} \ \log Q(\theta_0 + \epsilon)$$
$$\text{subject to} \ \epsilon^T g \leq y_\alpha \tag{9}$$

*coincides with the adjustment we make in SHAPO for the trust constraint $\delta_{Down} = \frac{z_\alpha^2}{2n}$, namely*
$\epsilon_\alpha = U_{\mathbf{F}}(-g, \frac{z_\alpha^2}{2n})$.

Intuitively, this result states that the adjusted parameter $\theta_0 + \epsilon_{\text{Down}}$ considered in *SHAPO* can be viewed as the most likely parameter under the parameter distribution $Q(\theta)$ that falls in the lower tail at the $\alpha$-quantile. Our approach *SHAPO*, which updates the current policy $\pi_0$ by using the gradient $\tilde{g}$ calculated at this adjusted parameter, can therefore be understood as promoting an enhancement of $\pi_0$ in a worst-case or pessimistic scenario, considering the existing epistemic uncertainty surrounding $\pi_0$. By applying *SHAPO* with a $\delta_{\text{Down}} > 0$ to update the actor, we are therefore effectively enforcing pessimism in face of epistemic uncertainty about our current policy.

Furthermore, by establishing the relationship $\delta_{\text{Down}} = \frac{z_\alpha^2}{2n}$, this proposition allows us to interpret the constraint $\delta_{\text{Down}}$ as a measure of confidence in the expected return of the current policy in the presence of epistemic uncertainty. More details and proof in Appendix B.

### 4.3 ANALYSIS OF *SHAPO* ON A SIMPLE GAUSSIAN POLICY

Our proposed approach to update the current policy $\pi_0$ is guided by the gradient $\tilde{g}$ of $L^\lambda_{\pi_0}(\theta)$ evaluated at $\theta_0 + \epsilon_{\text{Down}}$, while classical approaches use the gradient $g$ evaluated at $\theta_0$. Here we study how these two gradients $g$ and $\tilde{g}$ differ in a simplified setting.

We assume that the policy at some state is given by a $1D$ Gaussian policy $\pi(a; \mu, 1) = \mathcal{N}(a; \mu, 1)$ parametrized by the mean $\mu$ and that the current policy is $\pi_0(a) = \pi(a; 0, 1)$ specified by $\mu_0 = 0$. We propose to study the discrepancy between the two gradients $g$ and $\tilde{g}$ with respect to $\mu$ in terms of an observed action $a$ and an advantage value $A$ under a constraint $\delta_{\text{Down}}$. Our utility function $L^\lambda_{\pi_0}$ and its gradient can now be expressed as:

$$L(\mu; a, A) = \frac{A}{\pi_0(a)}\pi(a; \mu) \quad \nabla_\mu L(\mu; a, A) = \frac{A}{\pi_0(a)}\nabla_\mu \pi(a; \mu)$$

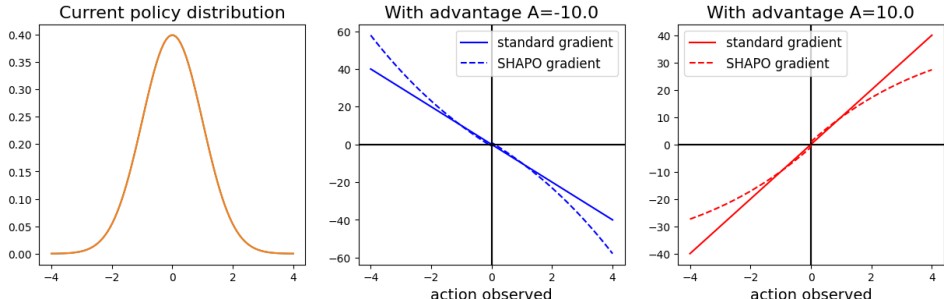

Figure 2: *1D Gaussian policy*: The gradient utilized in SHAPO exhibits a larger amplitude for rare and unsafe actions, while it is comparatively smaller for rare safe actions, compared to the classical gradient.

The standard gradient $g(a, A) = \nabla_\mu L(\mu; a, A)|_{\mu=0}$ and the gradient $\tilde{g}(a, A) = \nabla_\mu L(\mu; a, A)|_{\mu=\tilde{\mu}}$ at $\tilde{\mu} = \mu_0 + \epsilon_{\text{Down}}$ are as given in the following Proposition (See Appendix C for details).

**Proposition 3.** *Under the $1D$ Gaussian model, we have*

$$g(a, A) = aA \quad and \quad \tilde{g}(a, A) = A\Delta \exp\left(\frac{a^2 - \Delta^2}{2}\right) \quad with \quad \Delta = \begin{cases} a - \rho & if \ Aa < 0 \\ a + \rho & otherwise, \end{cases}$$

*where $\rho = \sqrt{2\delta_{Down}}$.*

Figure 2, show plots of the two gradients as functions of the action observed $a$ for two values of advantage ($A = -10$ and $A = 10$) with $\rho = 0.1$ (i.e. $\delta_{\text{Down}} = 0.005$).

We see that *SHAPO* gradients differ from standard gradients for rare actions $|a| > 1$. When the advantage is negative, indicating an unsafe action, we see that *SHAPO*'s gradient is larger compared to the standard gradient. In contrast, for positive advantage *SHAPO*'s gradient is smaller than the standard gradient. The analysis of this simplified setting shows that, under certain conditions, *SHAPO*'s policy update promotes a different treatment of rare actions depending on the sign of the corresponding advantage. When unsafe, rare actions are taken very seriously and an important update of the policy is promoted. In comparison, *SHAPO* opts for smaller updates when facing safe actions, even when they have large advantage. Overall, this behavior aligns well with our safe exploration objectives and helps explain why our approach favors policies that infrequently incur high costs (cf. Figure 5).

### 4.4 *SHAPO* FOR ON-POLICY RL ALGORITHMS

As presented in this section, Sharpness Aware Policy Optimization (*SHAPO*) modifies the TRPO algorithm with a Lagrange objective and pseudo-code is provided in Algorithm 3. We abstract this modification in Algorithm 1: this code returns a parameter direction $\tilde{g}$ for updating the current policy $\pi_\theta$ based on the objective function $L$, the Fisher matrix $F$ of the policy parametrization and constraint bound $\delta_{\text{Down}}$. Formulated this way, our method can be applied to most online policy RL algorithms and we provide pseudo-code in Appendix A for CRPO and CPO.

While Proposition 2 links the trust-region constraint $\delta_{\text{Down}}$ to the percentile $z_\alpha$ of the normal distribution (where $\alpha$ is the level of pessimism in face of uncertainty), the effective sample size $n$ cannot be easily estimated in continuous-control reinforcement learning where data are temporally correlated and only a subset of collected samples provide independent information. Furthermore, due to the complex coupling between the actor and critic, together with the challenges posed by exploration, it is difficult to determine the appropriate level of pessimism (*i.e.* the value of $\alpha$) to adopt at each stage of the training.

In this context, adopting a constant value of $\delta_{\text{Down}}$ during training and treating this value as tunable hyperparameter allows us to directly control the desired safety–efficiency tradeoff without relying on uncertain estimates of $n$. This makes the constant-$\delta_{\text{Down}}$ strategy both simple and effective, offering a stable mechanism for regulating pessimism throughout training. We further discuss this approach and other options to schedule the level of pessimism in Subsection 6.1 and in Appendix E.

Finally, in our method, we also incorporate Sharpness Aware Minimization with Euclidean neighborhoods for the critic, as proposed by Foret et al. (2020) for a supervised learning task. By considering

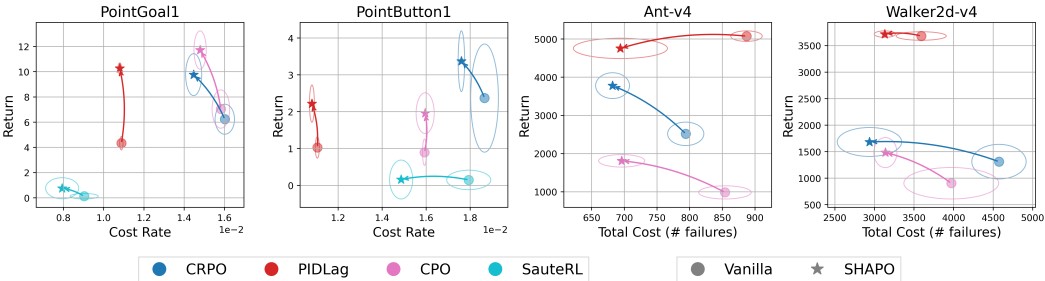

Figure 3: Performance of *SHAPO* on four different tasks across four different baselines over 5 training seeds. We report Cost Rate (Total Cost / Total Env Steps) for Safety Gym environments after training for $10M$ environment steps and Total Cost for MuJoCo environments after $5M$ environment steps. *SHAPO* (shown with a star) significantly improves the performance of the baseline algorithms (shown with a circle) on both safety and efficiency thus improving the safe exploration capabilities of the baseline algorithms. Width and height of the ellipses represent the standard error over 5 seeds.

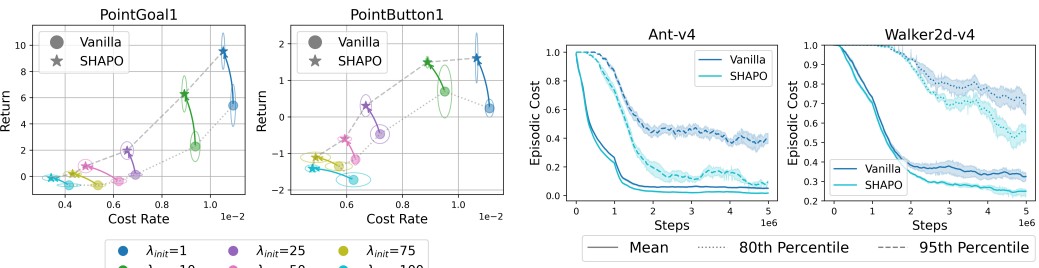

Figure 4: Results for different values of initial lambda ($\lambda_{init}$) for *PIDLag* algorithm with and without *SHAPO*. We can see that *SHAPO* improves upon the Vanilla *PIDLag* for all configurations.

Figure 5: *SHAPO* results in policies with episodic cost distributions with smaller tails (80th percentile and 95th percentile) crucially for safe exploration during the learning phase.

the sharpness of the loss landscape during the training of the critic, we aim to focus on flatter regions that yield more stable value function approximations.

## 5 EXPERIMENTAL SETUP

**Environments.** We evaluate *SHAPO* on Safety Gym (Ray et al., 2019) (*PointGoal1-v0*, *PointButton1-v0*) and MuJoCo (Todorov et al., 2012) (*Ant-v4*, *Walker2d-v4*). In *PointGoal1-v0*, an agent must reach a random goal while avoiding unsafe regions that accumulate cost; in *PointButton1-v0*, it must press buttons in the correct order, with penalties for wrong presses and additional hazards. In both tasks, the objective is to succeed while keeping cost under $\beta = 10$. In MuJoCo, the agent must run fast without falling, where each fall is a constraint violation.

**Baselines.** Although Sec. 4 more specifically describes how our approach modifies TRPO (Schulman et al., 2015) with a Lagrangian objective, *SHAPO* virtually applies to any on-policy method in a straightforward manner. We therefore also study the effect of *SHAPO* on CPO (Achiam et al., 2017), PIDLag (Stooke et al., 2020) (Lagrangian TRPO with PID control of $\lambda$), CRPO (Xu et al., 2021) (alternating primal updates) and SauteRL (Sootla et al., 2022) (state augmentation with constraint budget with TRPO (Schulman et al., 2015) as base algorithm). Pseudo-code for the *SHAPO* versions of the baseline algorithms is provided in Appendix A.

**Metrics.** In MuJoCo we report episodic return (efficiency) and total failures/cost (safety). In Safety Gym we track episodic return and cost during training, plus final average return and cost-rate (total cost / steps) as defined in (Ray et al., 2019). All results are generated by averaging over 5 seeds, the standard error in $x$ (Cost Rate / Total Cost) and $y$ (Return) are represented by the width and height of the ellipses, respectively.

**Implementation Details.** When adding *SHAPO*, baseline's hyperparameters are frozen and only $\delta_{\text{Down}}$ and $\rho_{critic}$ are tuned. Thus the safety–efficiency tradeoff is determined by the baseline, while *SHAPO* provides an orthogonal improvement. More details in the Appendix D.

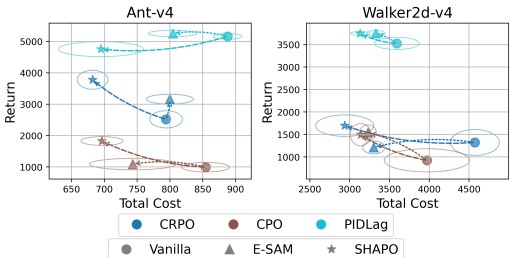 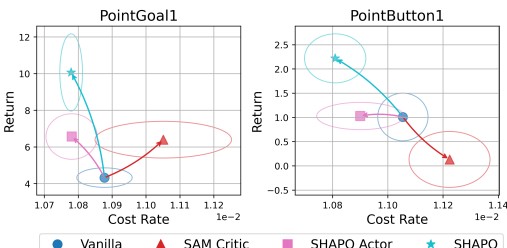

Figure 6: We compare results for *SHAPO* when using perturbation in the euclidean space *E-SAM* or in the Fisher Metric space (*SHAPO*). We can see that *SHAPO* is consistently better than *E-SAM* because of perturbation that ensure worsening of the inner objective.

Figure 7: *SHAPO* leads to significant improvement in cost-rate, even without SAM on critic. While SAM on critic leads to improvement in return, only adopting SAM for the critic does not promote safe exploration in *PointGoal1* and *PointButton1*

## 6  RESULTS

Fig. 3 summarizes performance across four baseline algorithms and their *SHAPO* counterparts on four tasks. In *PointGoal1* and *PointButton1*, we plot cost-rate (x-axis) against return (y-axis), capturing the safety–efficiency tradeoff during training. For *Ant-v4* and *Walker2d-v4*, we report cumulative failures (total cost). Across all benchmarks, *SHAPO* consistently improves either safety, task performance, or both. For instance, *SauteRL* (Sootla et al., 2022) achieves the lowest cost-rate on *PointGoal1* and *SHAPO* further reduces this cost without hurting returns. Since *SauteRL* relies on augmenting states with the remaining budget, it is inapplicable in MuJoCo (where $\beta = 0$ means any incurred cost violates the constraint), so we omit those results. Detailed results have been presented in the Appendix Fig. 14, 15, 16, 17.

The safety–efficiency tradeoff in Lagrangian methods is shaped by hyperparameters such as the initial multiplier $\lambda_{init}$. Fig. 4 shows results for *PIDLag* across different $\lambda_{init}$ values. In every setting, *SHAPO* expands the Pareto frontier of return versus cost, indicating that it consistently delivers better safety–efficiency tradeoffs than the vanilla algorithm.

To assess safety beyond mean statistics, Fig. 5 shows episodic cost distributions. While both vanilla baselines and *SHAPO* reduce average cost, only *SHAPO* consistently suppresses heavy tails ($95^{th}$ percentile in *Ant-v4*, $80^{th}$ in *Walker2d-v4*[2]). This tail suppression significantly enhances the agent's safety and aligns with our safe exploration objective. Consistent with our analysis in Sec. 4.2, *SHAPO*'s pessimism when faced with parameter uncertainty reduces catastrophic events, yielding fewer cumulative failures (i.e. Total Cost) (Fig. 3). Detailed plots are included in the Appendix.

### 6.1  ABLATIONS & ANALYSIS

*Fisher versus Euclidean perturbations:* Fig. 6 compares perturbations in Euclidean (E-SAM) and Fisher (*SHAPO*) spaces. *SHAPO* consistently outperforms E-SAM across tasks and baselines, validating that Fisher perturbations, being aligned with the local geometry of on-policy data, yield more principled and effective risk-sensitive updates.

*Actor versus critic perturbations:* Fig. 7 isolates the effect of applying *SHAPO* to different components. When SAM is applied only to the critic (SAM Critic), perturbations do not enforce risk-aversion and can even increase cost, since squared-error perturbations treat under- and overestimation symmetrically. In contrast, applying *SHAPO* only to the actor (*SHAPO* Actor) markedly improves safety by steering exploration conservatively. Finally, Using *SHAPO* (*SHAPO* Actor + SAM on critic) yields the strongest gains in both safety and efficiency.

*Pessimism schedule:* In light of Proposition 2, we consider for each update step $t$ the number $n_t$ of state-action pairs collected so far by our agent as a proxy for the effective sample size $n$ from Subsection 4.2. This allows us to interpret the bound $\delta_{\text{Down}}^t$ at step $t$ in terms of $n_t$ and the percentile $z_{\alpha_t}$ of the distribution $\mathcal{N}(0, 1)$ at the level of pessimism $\alpha_t$. Our simple strategy using a fixed $\delta_{\text{Down}}$ therefore can be interpreted as increasing the level of pessimism as the agent learns. Indeed, we then have $z_{\alpha_t} = -\sqrt{2n_t\delta_{\text{Down}}}$ and thus $\alpha_t = \Phi(-\sqrt{2n_t\delta_{\text{Down}}})$, where $\Phi$ is the cumulative distribution

---

[2]Walker2d-v4 is more challenging, with the $95^{th}$ percentile remaining near one, so we report the $80^{th}$ instead.

function of $\mathcal{N}(0,1)$. Another simple approach can instead consist of choosing a fixed value of pessimism $\alpha \in (0, \frac{1}{2})$ for the whole learning process, and then compute the bound $\delta^t_{\text{Down}} = \frac{z^2_\alpha}{2n_t}$ at step $t$, which effectively decreases as we progress through learning. Fig. 8 compares these two different approaches to scheduling pessimism. Figure 8 shows that both of these schedules allow *SHAPO* to consistently outperform the vanilla algorithm, highlighting the robustness of the proposed adjustment with respect to the choice of $\delta^t_{\text{Down}}$ which controls the level of pessimism. More details and analysis are provided in Appendix E.

## 7 RELATED WORK

*Uncertainty in RL:* RL agents must contend with aleatoric (outcome) and epistemic (model/parameter) uncertainty (Osband et al., 2016; Ghavamzadeh et al., 2015). Bayesian model-based methods propagate posterior uncertainty for exploration and risk assessment (Ghavamzadeh et al., 2015); model-free approaches use ensembles/randomized value functions for epistemic estimates (Osband et al., 2016; 2018; Mai et al., 2022) and distributional critics for risk-sensitive control (Bellemare et al., 2017; Dabney et al., 2018). Pessimistic objectives further bias learning toward conservative estimates (Kumar et al., 2020).

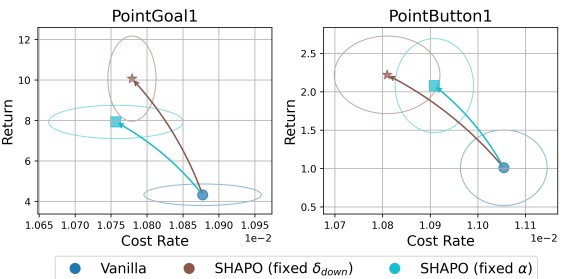

Figure 8: Pessimism schedule. We compare the selection of a constant $\delta_{\text{Down}}$ with the selection of a value $\alpha \in (0, 0.5)$ using the schedule $\delta^t_{\text{Down}} = \frac{z^2_\alpha}{2n_t}$ where $n_t$ is the number of state-action pairs seen by the agent. Both approaches outperform the vanilla approach.

*Sharpness-Aware Optimization:* Sharpness-aware objectives minimize the worst-case (or stochastic) loss in a local *noisy* parameter neighborhood (Rahn et al., 2023), yielding solutions that are insensitive to weight perturbations and thus more robust to parameter uncertainty (Foret et al., 2020; Andriushchenko & Flammarion, 2022; Möllenhoff & Khan, 2022). Geometry-aware variants (Kwon et al., 2021; Kim et al., 2022a) tailor the neighborhood to better approximate uncertainty directions, while noisy-neighborhood extensions explicitly inject randomness into the perturbation to probe loss under parameter noise and improve robustness (Baek et al., 2024). (Lee & Yoon, 2025) shows that applying SAM to PPO significantly improves robustness to aleatoric variations of the environment at test time. In this paper, we focus instead on the potential of sharpness aware optimization for safe exploration.

*Safe Exploration:* Safe exploration deals with epistemic uncertainty associated with data scarce regions of the state-space(Achiam et al., 2017; Liu et al., 2022; Sootla et al., 2022; Stooke et al., 2020; Mani et al., 2025; Gu et al., 2024). Strategies include Bayesian model-based control with uncertainty-aware planning (Berkenkamp, 2019), Lagrangian world models (Huang et al., 2023), safety critics that filter risky actions or impose constraints (Fisac et al., 2019; Srinivasan et al., 2020; Kang et al., 2022; Bharadhwaj et al., 2020), and state augmentation with accumulated cost as a risk proxy (Sootla et al., 2022; Jiang et al., 2023). In this paper, we are interested specifically in the epistemic uncertainty of the actor and in designing update rule to be pessimistic with respect to that uncertainty.

## 8 CONCLUSION

Safe exploration is ultimately about shaping updates to respect uncertainty, not only enforcing constraints on the final policy. By instantiating SAM in the actor's Fisher/KL geometry, we treat *sharpness*—the sensitivity of the policy surrogate to tiny parameter changes—as a practical proxy for epistemic uncertainty. When small nudges can flip outcomes, the local neighborhood is undersupported by data; *SHAPO* biases learning toward locally stable policies whose nearby variants behave similarly, tempering brittle, risk-seeking moves and emphasizing improvements that reliably reduce cost. This creates a closed loop: safer updates yield safer trajectories, which produce cleaner data in risky regions, reducing epistemic uncertainty and naturally relaxing conservatism as learning progresses.

## REPRODUCIBILITY STATEMENT

All code, configuration files, and exact hyperparameters required to reproduce our results are publicly available at `https://anonymous.4open.science/r/Safe-Policy-Optimization-813E`. Experiments were run in standard, publicly accessible environments (OpenAI Gym, Safety Gym), with version details documented in the repository.

## ACKNOWLEDGEMENTS

This work is supported by the DEEL Project CRDPJ 537462-18 funded by the Natural Sciences and Engineering Research Council of Canada (NSERC) and the Consortium for Research and Innovation in Aerospace in Québec (CRIAQ), together with its industrial partners Thales Canada inc, Bell Textron Canada Limited, CAE inc and Bombardier inc.[3]

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

---

**Algorithm 2** Sharpness-aware optimization

---

1: Initialize $\theta$
2: **while** not converged **do**
3:     Compute $\epsilon$ such that $\pi_{\theta+\epsilon} \approx \arg\min_{\tilde{\pi} \in N(\pi)} \mathcal{L}(\tilde{\pi})$
4:     Calculate the gradient at perturbed parameter: $g \leftarrow \nabla \mathcal{L}(\theta + \epsilon)$
5:     Update parameters: $\theta \leftarrow \theta + \alpha g$ {$\alpha$ is the learning rate}
6: **end while**

---

**Algorithm 3** *SHAPO* (TRPOLag): Sharpness Aware Policy Optimization (TRPOLag)

---

1: Initialize parameters $\theta$
2: Set trust region constraint $\delta_{\text{Up}}$ and *SHAPO* constraint $\delta_{\text{Down}}$
3: **while** not converged **do**
4:     Compute the Fisher matrix $F$ at $\theta$
5:     Compute the gradient: $g \leftarrow \nabla_\theta L_{\pi_\theta}^\lambda(\theta)|_\theta$
6:     Compute the perturbation: $\epsilon_{\text{Down}} = U_F(-g, \delta_{\text{Down}})$
7:     Compute the gradient at perturbed parameter: $\tilde{g} \leftarrow \nabla_\theta L_{\pi_\theta}^\lambda(\theta)|_{\theta+\epsilon_{\text{Down}}}$
8:     Compute the parameter update: $\epsilon_{\text{Up}} = U_F(\tilde{g}, \delta_{\text{Up}})$
9:     Update parameters: $\theta \leftarrow \theta + \epsilon_{\text{Up}}$
10: **end while**
11: Return policy $\pi_\theta$

---

**Algorithm 4** *SHAPO* (PPO): Sharpness-Aware Policy Optimization (PPO)

---

1: Initialize policy parameters $\theta$
2: Set clipping parameter $\epsilon$ and *SHAPO* constraint $\delta_{\text{Down}}$
3: Set learning rate $\eta$
4: **while** not converged **do**
5:     Compute the Fisher matrix F at $\theta$
6:     Compute SHAPO gradient for the clipped objective: $\tilde{g} \leftarrow SHAPO(\theta, L_{\pi_\theta}^{\text{PPO}}, F, \delta_{down})$
7:     Update parameters with the *SHAPO* gradient: $\theta \leftarrow \theta + \eta\, \tilde{g}$.
8: **end while**

---

# A   Pseudo-code

Pseudo-code for Sharpness Aware Optimization is presented in Algorithm 2 and for our Sharpness Aware Policy Optimization for Lagrangian TRPO (*SHAPO* TRPOLag) in Algorithm 3.

Recall that $U_F(g, \delta)$ stands for the (estimated) solution to the optimization problem 5

$$U_F(g, \delta) = \arg\max_{\frac{1}{2}\epsilon^T F\epsilon \leq \delta} \langle g, \epsilon \rangle. \tag{10}$$

where $\delta > 0$, $g$ is a vector and $F$ is a symmetric positive definite matrix. In TRPO at current $\theta_0$, $g$ is the gradient of $L_{\pi_{\theta_0}}(\theta)$ evaluated at $\theta_0$ and $F$ is the Fisher information matrix $\mathbf{F}_{\theta_0}$ of the policy parametrization $\pi_\theta, \theta \in \Theta$, evaluated at $\theta_0$.

We provide pseudo-code for *SHAPO* in Algorithm 1. Formulated this way, our method can be applied to most on-policy RL algorithms. It returns a parameter direction $\tilde{g}$ for updating the current policy $\pi_\theta$ based on the objective function, the Fisher matrix of the policy parametrization and constraint bound $\delta_{\text{Down}}$. This function effectively abstracts lines $4 - 7$ of Algorithm 3 and allows us to present how our method applies to other algorithms.

While TRPOLag combines cost and reward into a Lagrangian objective $L_{\pi_\theta}^\lambda$ at the current policy $\pi_0$ (cf. Equation 3), other algorithms like CPO or CRPO consider two distinct objectives. Given the

---

**Algorithm 5** *SHAPO*-CRPO

---
1: Initialize parameters $\theta$
2: Set trust region constraint $\delta_{\text{Up}}$ and *SHAPO* constraint $\delta_{\text{Down}}$
3: **while** not converged **do**
4:     Compute the Fisher matrix $F$ at $\theta$
5:     **if** Expected Cost > constraint threshold ($\beta$) **then**
6:         Compute SHAPO gradient on Cost Surrogate $\tilde{g} \leftarrow SHAPO(\theta, L^c_{\pi_\theta}, F, \delta_{\text{Down}})$
7:     **else**
8:         Compute SHAPO gradient on Reward Surrogate $\tilde{g} \leftarrow SHAPO(\theta, L^r_{\pi_\theta}, F, \delta_{\text{Down}})$
9:     **end if**
10:     Compute the parameter update: $\epsilon_{\text{Up}} = U_F(\tilde{g}, \delta_{\text{Up}})$
11:     Update parameters: $\theta \leftarrow \theta + \epsilon_{\text{Up}}$
12: **end while**
13: Return policy $\pi_\theta$

---

**Algorithm 6** *SHAPO*-CPO

---
1: Initialize parameters $\theta$
2: Set trust region constraint $\delta_{\text{Up}}$ and *SHAPO* constraint $\delta_{\text{Down}}$
3: **while** not converged **do**
4:     Compute Fisher matrix $F$ at $\theta$
5:     Compute SHAPO gradient on Cost Surrogate $\tilde{g}_r \leftarrow SHAPO(\theta, L^r_{\pi_\theta}, F, \delta_{\text{Down}})$
6:     Compute SHAPO gradient on Reward Surrogate $\tilde{g}_c \leftarrow SHAPO(\theta, L^c_{\pi_\theta}, F, \delta_{\text{Down}})$
7:     Resolve gradient update $\epsilon_{\text{Up}} \leftarrow CPO\_Update(\tilde{g}_r, \tilde{g}_c, \delta_{\text{Up}})$
8:     Update parameters: $\theta \leftarrow \theta + \epsilon_{\text{Up}}$
9: **end while**
10: Return policy $\pi_\theta$

---

current policy $\pi_0$ we define the two following objectives similarly to Equation 3:

$$L^r_{\pi_0}(\theta) = \mathbb{E}_{s \sim d^0} \mathbb{E}_{a \sim \pi_0(a|s)} \frac{A^0_r(s,a)}{\pi_0(a|s)} \pi_\theta(a|s) \tag{11}$$

$$L^c_{\pi_0}(\theta) = \mathbb{E}_{s \sim d^0} \mathbb{E}_{a \sim \pi_0(a|s)} \frac{A^0_c(s,a)}{\pi_0(a|s)} \pi_\theta(a|s), \tag{12}$$

where $d^0$ denotes the discounted state distribution under $\pi_0$ and $A^0_r(a,s)$ (resp.$A^0_c$) quantifies the advantage in reward (resp. cost) of taking action $a$ in state $s$ compared to the average action under $\pi_0$ in state $s$. More precisely, letting $Q^0_r(s,a) = E_{\tau \sim \pi_0}[R_r(\tau)|s_0 = s, a_0 = a]$ for the reward action-value function associated with $\pi_0$, where $R_r(\tau) = \sum_{t=0}^\infty \gamma^t r(s_t, a_t)$, we have $A^0_\lambda(s,a) = Q^0_r(s,a) - E_{a \sim \pi_0(\ |s)}[Q^0_r(s,a)]$ and similarly for cost with $R_c(\tau) = \sum_{t=0}^\infty \gamma^t c(s_t, a_t)$.

While CRPO's update relies on a single gradient direction for either improving reward or decreasing cost (see Algorithm 5) at each step, CPO relies on both pieces of information through an update function that we denote by *CPO_Update* (see Algorithm 6). In both cases, our method *SHAPO* provides gradient directions for both reward and cost following the same steps, thereby incorporating pessimism in face of epistemic uncertainty of the actor into these algorithms. Since SauteRL (Sootla et al., 2022) is a state-augmentation technique, it can use any underlying policy gradient algorithm. We've used TRPO (Schulman et al., 2015) as the base algorithm for SauteRL and the *SHAPO* version can be derived from Algorithm 3. In Algorithm 4, we propose a simple approach to apply *SHAPO* on the popular PPO algorithm that uses the following clipped objective (for some $\epsilon > 0$):

$$L^{\text{PPO}}_{\pi_0}(\theta) = \mathbb{E}_{s \sim d^0} \mathbb{E}_{a \sim \pi_0(a|s)} \left[ \min \left( r(\theta, a, s) A^0(s,a), \text{clip}(r(\theta, a, s), 1 - \epsilon, 1 + \epsilon) A^0(s,a) \right) \right],$$

where $r(\theta, a, s) = \frac{\pi_\theta(a|s)}{\pi_0(a|s)}$.

## B REINTERPRETING FISHER SAM AS PESSIMISM IN FACE OF EPISTEMIC UNCERTAINTY

For the sake of completeness, we prove first the following well known result (cf. Kakade (2001); Schulman et al. (2015)):

**Lemma 4.** *The solution to the optimization problem*

$$U_F(g, \delta) = \arg\max_{\frac{1}{2}\epsilon^T F\epsilon \leq \delta} \langle g, \epsilon \rangle \tag{13}$$

*is given by $U_F(g, \delta) = \frac{\sqrt{2\delta}}{\sqrt{g^T F^{-1} g}} F^{-1} g$ where $F^{-1}$ is the inverse of $F$, which is assumed to be symmetric positive definite.*

*Proof.* We introduce a Lagrange multiplier $\lambda$ for the constraint $\frac{1}{2}\epsilon^T F\epsilon \leq \delta$ and define the following Lagrangian:

$$\mathcal{L}(\epsilon, \lambda) = \langle g, \epsilon \rangle - \lambda \Big(\frac{1}{2}\epsilon^T F\epsilon - \delta\Big).$$

We then find the stationary point $\mathcal{L}(\epsilon, \lambda)$. First, we have:

$$0 = \nabla_\epsilon \mathcal{L}(\epsilon, \lambda) = g - \lambda F\epsilon$$

which leads to $\epsilon = \frac{1}{\lambda} F^{-1} g$. Next we have:

$$0 = \nabla_\lambda \mathcal{L}(\epsilon, \lambda) = \frac{1}{2}\epsilon^T F\epsilon - \delta,$$

that leads to the constraint $\frac{1}{2}\epsilon^T F\epsilon = \delta$. Substituting $\epsilon = \frac{1}{\lambda} F^{-1} g$ into the constraint, and using the fact that $F^{-1}$ is also symmetric, we find:

$$\frac{1}{2\lambda^2} g^T F^{-1} g = \delta \quad \Rightarrow \quad \lambda = \sqrt{\frac{g^T F^{-1} g}{2\delta}}$$

Consequently, as desired:

$$\epsilon = \sqrt{\frac{2\delta}{g^T F^{-1} g}} F^{-1} g.$$

$$\square$$

Recall that $Q(\theta)$ is assumed to follow a multivariate normal distribution $\mathcal{N}(\theta_0, \frac{1}{n} F^{-1})$ where $F^{-1}$ is the inverse of the Fisher matrix at $\theta_0$. Recall that $Y = g^t(\theta - \theta_0)$ with $\theta \sim Q$ with $\alpha$-quantiles denoted by $y_\alpha$. Finally $z_\alpha$, denotes the $\alpha$-quantile for $\mathcal{N}(0, 1)$. We now provide the proof of Proposition 2 which is illustrated in Figure 1:

**Proposition 5.** *Let $Q(\theta) = \mathcal{N}(\theta_0, \frac{1}{n} F^{-1})$. For every $\alpha \in (0, \frac{1}{2})$ the solution $\epsilon_\alpha$ to the optimization problem*

$$\underset{\epsilon}{maximize} \ \log Q(\theta_0 + \epsilon) \tag{14}$$
$$subject \ to \ g^T \epsilon \leq y_\alpha$$

*coincides with the adjustment we make in SHAPO, namely we have $\epsilon_{Down} = U_F(-g, \delta_{Down})$ when $\delta_{Down} = \frac{z_\alpha^2}{2n}$.*

*Proof.* Note that the precision matrix of $Q(\theta)$ is equal to $nF$ and so

$$\log Q(\theta_0 + \epsilon) = -\frac{n}{2}\epsilon^T F\epsilon + \text{constant}.$$

We introduce a Lagrange multiplier $\lambda$ for the constraint $g^T \epsilon \leq y_\alpha$ and define the following Lagrangian:

$$\mathcal{L}(\epsilon, \lambda) = -\frac{n}{2}\epsilon^T F\epsilon - \lambda\big(g^T \epsilon - y_\alpha\big).$$

Next, we look for the stationary points of $\mathcal{L}$. First,

$$0 = \nabla_\epsilon \mathcal{L}(\epsilon, \lambda) = -nF\epsilon - \lambda g$$

which implies $\epsilon = -\frac{\lambda}{n}F^{-1}g$. Next we see that

$$0 = \nabla_\lambda \mathcal{L}(\epsilon, \lambda) \quad \Rightarrow \quad g^T \epsilon = y_\alpha.$$

Therefore,

$$y_\alpha = \frac{\lambda}{n} g^T F^{-1} g \quad \text{and so} \quad \lambda = \frac{n y_\alpha}{g^T F^{-1} g}.$$

Consequently, we obtain

$$\epsilon = -\frac{y_\alpha}{g^T F^{-1} g} F^{-1} g.$$

Now using the fact that $y_\alpha = \sigma z_\alpha$ where $\sigma = \frac{1}{\sqrt{n}}\sqrt{g^T F^{-1} g}$ is the standard deviation of $Y = g^t \epsilon$ with $\epsilon \sim Q(\theta)$, we get

$$\epsilon = -\frac{z_\alpha}{\sqrt{n g^T F^{-1} g}} F^{-1} g.$$

By Lemma 4, we have

$$U_F(-g, \delta) = -\sqrt{\frac{2\delta}{g^T F^{-1} g}} F^{-1} g,$$

which is equal to $\epsilon$ when $\frac{z_\alpha}{\sqrt{n}} = \sqrt{2\delta}$, namely when $\delta = \frac{z_\alpha^2}{2n}$, as desired. $\qquad\square$

We observe that the relation $\delta_{\text{Down}} = \frac{z_\alpha^2}{2n}$ offers an attractive perspective on the trust constraint for the inner minimization in *SHAPO*. This equation expresses the bound in terms of the $\alpha$-quantile of the standard normal distribution $\mathcal{N}(0, 1)$ and $n$, which represents the sample size used for estimating $\theta_0$. For a given sample size $n$, selecting a confidence level—such as the 5th percentile—yields a specific value for $\delta_{\text{Down}}$. This approach implies that one can initially set $\delta_{\text{Down}}^{\text{init}} = z_\alpha^2$ by determining an appropriate confidence level $\alpha \in (0, \frac{1}{2})$. As the actor gains more experience, it would appear recommended to decrease $\delta_{\text{Down}}$ according to $\delta_{\text{Down}} = \frac{\delta_{\text{Down}}^{\text{init}}}{2n}$, reflecting the increasing trust in the estimates as more data becomes available. This strategy not only enhances the robustness of the minimization process but also aligns the trust constraint with the actor's growing experience.

## C   ANALYSIS OF SHAPO ON A SIMPLE GAUSSIAN POLICY

Our proposed approach to update the current policy $\pi_0$ is guided by the gradient $\tilde{g}$ of $L_{\pi_0}^\lambda(\theta)$ evaluated at $\theta_0 + \epsilon_{\text{Down}}$, while classical approaches use the gradient $g$ evaluated at $\theta_0$. Here we study how these two gradients $g$ and $\tilde{g}$ differ in a simplified setting. We observe that the contribution of a single state-action pair $(s, a)$ towards $L_{\pi_0}^\lambda$ is given by:

$$L_{\pi_0}^\lambda(\theta|s, a) = \frac{A^0(s, a)}{\pi_0(a|s)} \pi_\theta(a|s),$$

whose gradient with respect to $\theta$ is given by

$$\nabla_\theta L_{\pi_0}^\lambda(\theta|s, a) = \frac{A^0(s, a)}{\pi_0(a|s)} \nabla_\theta \pi_\theta(a|s)$$

Each observed state-action pair therefore contributes towards the gradient, to increase or decrease the likelihood of action $a$ depending on the sign of $A^0(s, a)$ (in way that is inversely proportional to the likelihood of that action under $\pi_0$).

Here we study how the two gradients $g$ (standard) and $\tilde{g}$ (*SHAPO*) differ in a simplified setting. We assume that the policy at some state is given by a $1D$ Gaussian policy $\pi(a; \mu, 1) = \mathcal{N}(a; \mu, 1)$ parametrized by the mean $\mu$, with probability density function given by:

$$\pi(a; \mu) = \frac{1}{\sqrt{2\pi}} \exp\left(-\frac{(a - \mu)^2}{2}\right).$$

The gradient of $\pi(a; \mu)$ with respect to $\mu$ is expressed as:

$$\frac{\partial}{\partial \mu}\pi(a;\mu) = (a-\mu)\pi(a;\mu)$$

Assuming that the current policy is $\pi_0(a) = \pi(a; 0, 1)$ specified by $\mu_0 = 0$, we study the discrepancy between the two gradients $g$ and $\tilde{g}$ with respect to $\mu$ in terms of an observed action $a$ and an advantage value $A$. Our utility function $L_{\pi_0}^\lambda$ and its gradient can now be expressed as:

$$L_{\pi_0}^\lambda(\mu; a, A) = \frac{A}{\pi_0(a)}\pi(a;\mu) \quad \nabla_\mu L_{\pi_0}^\lambda(\mu; a, A) = \frac{A}{\pi_0(a)}\nabla_\mu\pi(a;\mu)$$

The gradient $g(a, A)$ at $\mu_0$ becomes:

$$g(a, A) = \nabla_\mu L_{\pi_0}^\lambda(\mu|a)|_{\mu=0} = \frac{A}{\pi_0(a)}(a-\mu_0)\pi(a;\mu_0) = Aa$$

Therefore, in this situation, the standard gradient is simply the product of the advantage $A$ with the action $a$. With *SHAPO* the gradient is instead computed at $\mu_0 + \epsilon_{\text{Down}}$, where $\epsilon_{\text{Down}}$ is a parameter perturbation crafted to minimize $L_{\pi_0}^\lambda$. Note that with our choice of parametrization $\text{KL}(\pi(\ ;\mu_0,1)\|\pi(\ ;\mu_0 + \epsilon_{\text{Down}},1)) = \frac{\epsilon_{\text{Down}}^2}{2}$, which is half the square of the Euclidean distance between mean parameters. Therefore, the approach based on the Kullback-Leibler neighborhood $(\text{KL}(\pi(\ ;\mu_0,1)\|\pi(\ ;\mu_0 + \epsilon_{\text{Down}},1) \leq \delta_{\text{Down}})$ coincides with the one using the Euclidean metric on parameters $(\epsilon_{\text{Down}} \leq \rho)$ and can be expressed in terms of $\rho = \sqrt{2\delta_{\text{Down}}}$. The perturbation is therefore given by:

$$\epsilon_{\text{Down}}(a, A) = -\rho\frac{g(a, A)}{|g(a, A)|}$$

This adjustment contributes to increase the likelihood of action $a$ when the advantage $A$ is negative (unsafe action) and to decrease it when $A$ is positive. The adjusted parameter $\tilde{\mu}(a, A) = \mu_0 + \epsilon_{\text{Down}}(a, A)$ is given by:

$$\tilde{\mu}(a, A) = -\rho\,\text{sign}(Aa)$$

So, for example, this adjusted mean is negative when $A < 0$ and $a < 0$, thereby leading to a likelihood $\pi(a; \tilde{\mu}, 1)$ larger than $\pi(a; 0, 1)$.

Now we compute the gradient $\tilde{g}$ at $\tilde{\mu}$:

$$\tilde{g}(a, A) = \nabla_\mu L_{\pi_0}^\lambda(\mu|a)|_{\mu=0} = \frac{A}{\pi_0(a)}(a-\tilde{\mu})\pi(a;\tilde{\mu})$$

As desired, this leads to the following expression for $\tilde{g}(a, A)$

$$\tilde{g}(a, A) = \begin{cases} A(a-\rho)\exp\left(\frac{a^2-(a-\rho)^2}{2}\right) & \text{if } Aa < 0 \\ A(a+\rho)\exp\left(\frac{a^2-(a+\rho)^2}{2}\right) & \text{if } Aa \geq 0. \end{cases}$$

## D  IMPLEMENTATION DETAILS & ANALYSIS

### D.1  HYPERPARAMETER DETAILS

For the base algorithms we use the optimized hyperparameters available in open-source implementations like Omnisafe (Ji et al., 2024)[4]. *SHAPO* hyperparameters $\delta_{\text{Down}}$ and $\rho_{critic}$ are optimized by performing grid search between range $[0.001, 0.00001]$ for $\delta_{\text{Down}}$ and $\{0.01, 0.05, 0.005\}$ for $\rho_{critic}$ while keeping the hyperparameters of the base algorithm were kept fixed. Full list of hyperparameters for each baseline and the corresponding environment have been provided here (https://anonymous.4open.science/r/Safe-Policy-Optimization-813E). Most common choice of *SHAPO* hyperparameters that gave the best performance across tasks and baselines

---

[4]https://github.com/PKU-Alignment/omnisafe

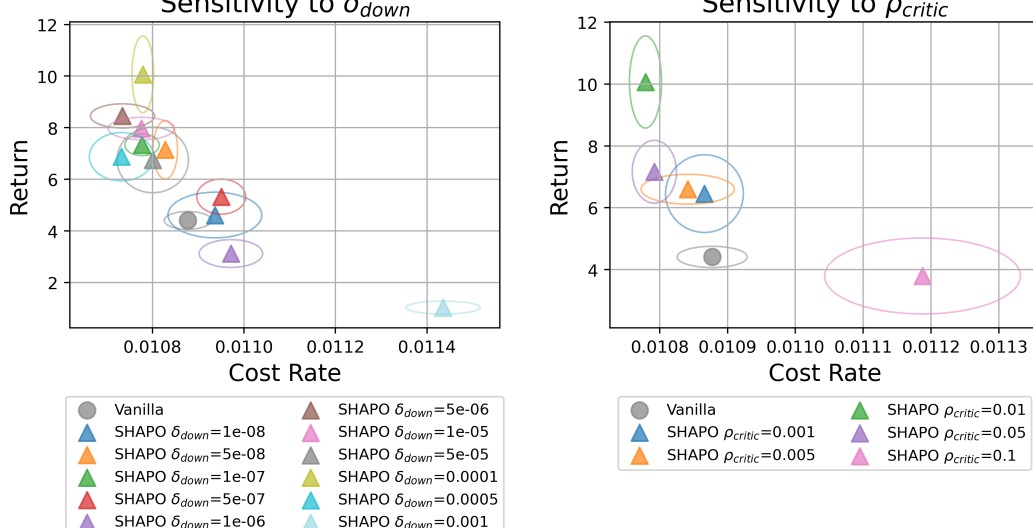

Figure 9: *Hyperparameter Sensitivity:* We evaluate the sensitivity of *SHAPO* to $\delta_{\text{Down}}$ (left) and $\rho_{critic}$ (right). Very small $\delta_{\text{Down}}$ effectively disables *SHAPO* and yields performance similar to the vanilla baseline, while overly large values destabilize training. Moderate values provide the most consistent improvements. For $\rho_{critic}$, performance is stable across a broad range, as long as the value is not excessively large.

was $\delta_{\text{Down}} = 0.0001$ and $\rho_{critic} = 0.01$. There is a very clear monotonous hyperparameter sensitivity to *SHAPO* i.e. if the $\delta_{\text{Down}}$ value is too small, the policy optimization is unchanged from the base algorithm thus the performance also resembles the base algorithm's performance, while if the $\delta_{\text{Down}}$ is too high, the state-distributions of the original policy $d^{\pi_0}$ that generated the data and that of perturbed policy $d^{\tilde{\pi}}$ are no longer bounded and thus its no longer possible to provide a tight bound on the deterioration of the policy required for the inner SHAPO objective, which leads to unstability and inferior performance.

## D.2 HYPERPARAMETER SENSITIVITY

To study the robustness of *SHAPO* to the choice of hyperparameters, we conduct an analysis (Fig. 9) varying the two key hyperparameters, $\delta_{down}$ and $\rho_{critic}$, while keeping all other settings fixed. This setup allows us to isolate the effect of each hyperparameter on safety and efficiency of the resulting policy. The results show that very small $\delta_{down}$ values effectively disable the perturbation step in *SHAPO*, causing the algorithm to behave like the vanilla baseline. Conversely, excessively large $\delta_{down}$ values lead to unstable updates and degraded performance. We observe a clear band of intermediate values that con-

| Metric | Vanilla | *SHAPO* |
|---|---|---|
| Total Runtime (mins) | 85 min | 100 min |
| Update Time (s) | 4 s | 5.5 s |

Table 1: Table shows the total runtime of Vanilla (*PIDLag*) and *SHAPO* (*PIDLag*) on *PointGoal1* task as well as time taken for individual updates. As we can see that the compute overhead is negligible in comparision to the overall time it takes to train the RL agent and most of the runtime is taken up by collecting data through policy rollouts.

sistently improves both learning speed and final performance. In contrast, $\rho_{critic}$ exhibits much less sensitivity: performance remains stable across a wide range of values, with degradation only occurring when the regularization becomes extremely large. These trends confirm that *SHAPO* is robust to reasonable choices of $\rho_{critic}$ and requires only mild tuning of $\delta_{down}$ to achieve reliable gains.

## D.3 RUNTIME ANALYSIS

Here we analyze the computational overhead introduced by *SHAPO* during training across different tasks. In our experiments, overall training time is dominated by environment interaction rather than

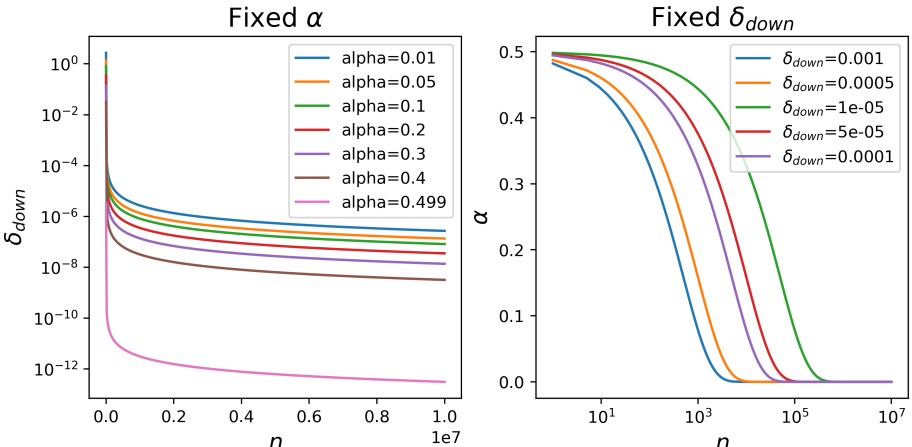

Figure 10: Comparison of pessimism schedules under *SHAPO*. **Left: Fixed** $\alpha$**.** Holding the pessimism level $\alpha$ constant causes the corresponding bound $\delta_{\text{Down}}^t = z_\alpha^2/(2n_t)$ to shrink rapidly as $n_t$ grows, reducing the robustness margin over time. **Right: Fixed** $\delta_{\text{Down}}$**.** Keeping $\delta_{\text{Down}}$ constant makes the implied percentile $\alpha_t = \Phi(-\sqrt{2n_t \delta_{\text{Down}}})$ decrease with $n_t$, meaning the agent becomes more risk-averse as training progresses. This is desirable for safe exploration: early on, when data are limited, pessimism remains low to avoid overly conservative behavior, while increased pessimism later helps guard against epistemic uncertainty once sufficient experience has been gathered.

by policy or critic updates. Consequently, the additional TRPO step required for *SHAPO*'s parameter adjustment contributes only marginally to the total wall-clock time, which includes data collection, policy updates, and critic updates. Table 1 reports the runtime for training an RL agent for 10M timesteps on the *PointGoal1* task averaged across 5 different training runs. As shown, *SHAPO* incurs only a small increase in total runtime, since most of the compute is spent on collecting on-policy rollouts. The per-update computation time for the actor and critics increases by roughly 30 percent, but this overhead is minor compared to the dominant cost of data collection. All the experiments were performed on CPU with 10 cores and 30 GB of RAM. To increase the efficiency of data-collection, we use vectorized environments provided with gymnasium (Towers et al., 2024) and safety-gymnasium(Ray et al., 2019), with 5 environments running in parallel.

## E    PESSIMISM SCHEDULE

In order to compare different ways of controlling the level of pessimism used in *SHAPO*, we use at each update step $t$ the number of collected state–action samples $n_t$ as a proxy for the ideal sample size $n$ introduced in Subsection 4.2. It is important to note that the true value of $n$—the number of *informative, approximately independent* samples governing posterior contraction—is unknown in practice. The proxy $n_t$ may therefore increase much faster than the effective statistical sample size, which has direct implications for how pessimism evolves during learning.

Increasing $n_t$ affects the pessimism level differently depending on whether we fix $\delta_{\text{Down}}$ or fix the target percentile $\alpha$. When $\delta_{\text{Down}}$ is kept constant, Proposition 2 implies that the induced percentile satisfies

$$z_\alpha^t = -\sqrt{2n_t \delta_{\text{Down}}}, \qquad \alpha_t = \Phi(z_\alpha^t),$$

so that $\alpha_t$ decreases as $n_t$ grows. Thus, the agent becomes *increasingly pessimistic*, focusing on more extreme tail events as training progresses. This behavior is desirable for safe exploration: early in learning, when data are scarce and uncertainty is large, low pessimism avoids overly conservative behavior that would suppress exploration; as more experience is gathered, increasing pessimism helps guard against refined estimates of unsafe regions.

By contrast, if we fix a pessimism level $\alpha \in (0, \frac{1}{2})$, then the corresponding $\delta_{\text{Down}}^t$

$$\delta_{\text{Down}}^t = \frac{z_\alpha^2}{2n_t}$$

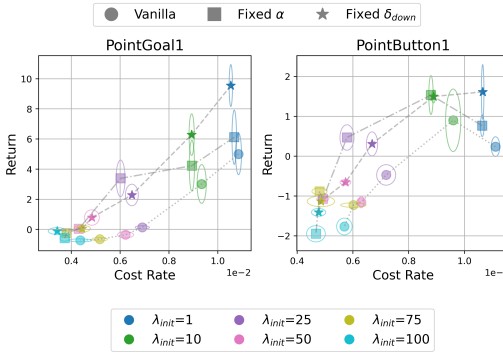

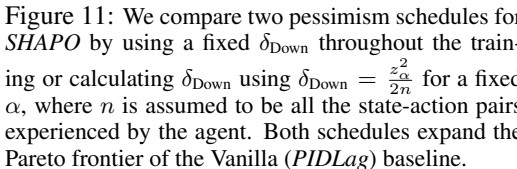

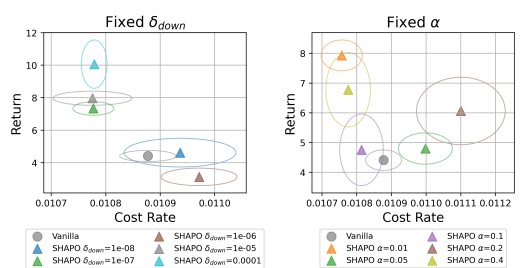

Figure 11: We compare two pessimism schedules for *SHAPO* by using a fixed $\delta_{\text{Down}}$ throughout the training or calculating $\delta_{\text{Down}}$ using $\delta_{\text{Down}} = \frac{z_\alpha^2}{2n}$ for a fixed $\alpha$, where $n$ is assumed to be all the state-action pairs experienced by the agent. Both schedules expand the Pareto frontier of the Vanilla (*PIDLag*) baseline.

Figure 12: Both fixed $\delta_{\text{Down}}$ and fixed $\alpha$ schedules lead to significant improvement in performance over the vanilla baseline on *PointGoal1* task. Although the choice of $\alpha$ is not straightforward as both very low ($\alpha = 0.01$) and high alphas ($\alpha = 0.4$) give good performance.

decays at rate $1/n_t$. Because $n_t$ may grow much faster than the true effective sample size, this schedule can drive $\delta_{\text{Down}}^t$ to zero prematurely, effectively nullifying any pessimism in the later stages of learning. This mirrors the Bernstein–von Mises posterior contraction, but may not model appropriately the epistemic uncertainty state of an actor trained in a reinforcement learning setting, where epistemic uncertainty persists due to partial observability, non-stationarity, or limited coverage of the state space. Fig. 10 illustrates this phenomenon: the fixed-$\alpha$ schedule quickly collapses the $\delta_{\text{Down}}$ as $n_t$ grows, whereas fixing $\delta_{\text{Down}}$ maintains a nontrivial degree of pessimism throughout training.

Fig. 11 and 12 compare these two approaches to scheduling pessimism. Both strategies—fixing $\delta_{\text{Down}}$ or fixing $\alpha$—enable *SHAPO* to outperform the vanilla baseline (*PIDLag* in this case). Fig. 11 shows that both pessimism schedules expand the Pareto frontier of the vanilla algorithm enhancing safety-efficiency tradeoff across different $\lambda_{init}$ values. Fig. 12 compares the performance of *SHAPO* compared to the vanilla baseline (in grey) for different choices of $\delta_{\text{Down}}$ or $\alpha$. We can see that there are many choices of fixed $\alpha$'s and $\delta_{\text{Down}}$'s that lead to improvement over the vanilla *PIDLag*, but the choice of $\alpha$ that leads to best performance is not that obvious as it impacts the exploration of the algorithm.

Since the effective sample size $n$ cannot be reliably estimated in continuous-control reinforcement learning—where data are temporally correlated and only a fraction of collected samples contribute independent information—fixing $\delta_{\text{Down}}$ provides a practical and robust alternative. Treating $\delta_{\text{Down}}$ as a tunable hyperparameter allows practitioners to directly control the desired safety–efficiency tradeoff without relying on uncertain estimates of $n$. This makes the fixed-$\delta_{\text{Down}}$ strategy both simple and effective, offering a stable mechanism for regulating pessimism throughout training.

# F  *SHAPO* FOR UNCONSTRAINED RL

*SHAPO* is a general mechanism that can be incorporated into any on-policy reinforcement learning algorithm, and its utility extends beyond explicitly constrained RL settings. To illustrate this versatility, we evaluate *SHAPO* in three standard MuJoCo (Todorov et al., 2012) continuous-control environments: *Ant-v4*, *Walker2d-v4*, and *HalfCheetah-v4*. The first two environments (*Ant* and *Walker2d*) include termination conditions in which the episode ends immediately if the agent falls. Although they do not provide an explicit cost signal or negative penalty for falling, this termination logic implicitly induces a lower tail of poor outcomes: each fall truncates the episode, prevents further reward collection, and therefore constitutes an undesirable event in the return distribution. In contrast, *HalfCheetah* has no such termination or failure mode—the episode continues regardless of how the agent moves—yielding a much smoother return distribution without a pronounced lower tail. We compare PPO (Schulman et al., 2017) to its *SHAPO*-augmented variant (*SHAPO*-PPO), implemented as described in Algorithm 4. From this perspective, *SHAPO*'s role becomes clear: it modulates the actor's sensitivity to tail events in the loss landscape, whether these arise from explicit constraints or from inherent structural properties of the environment.

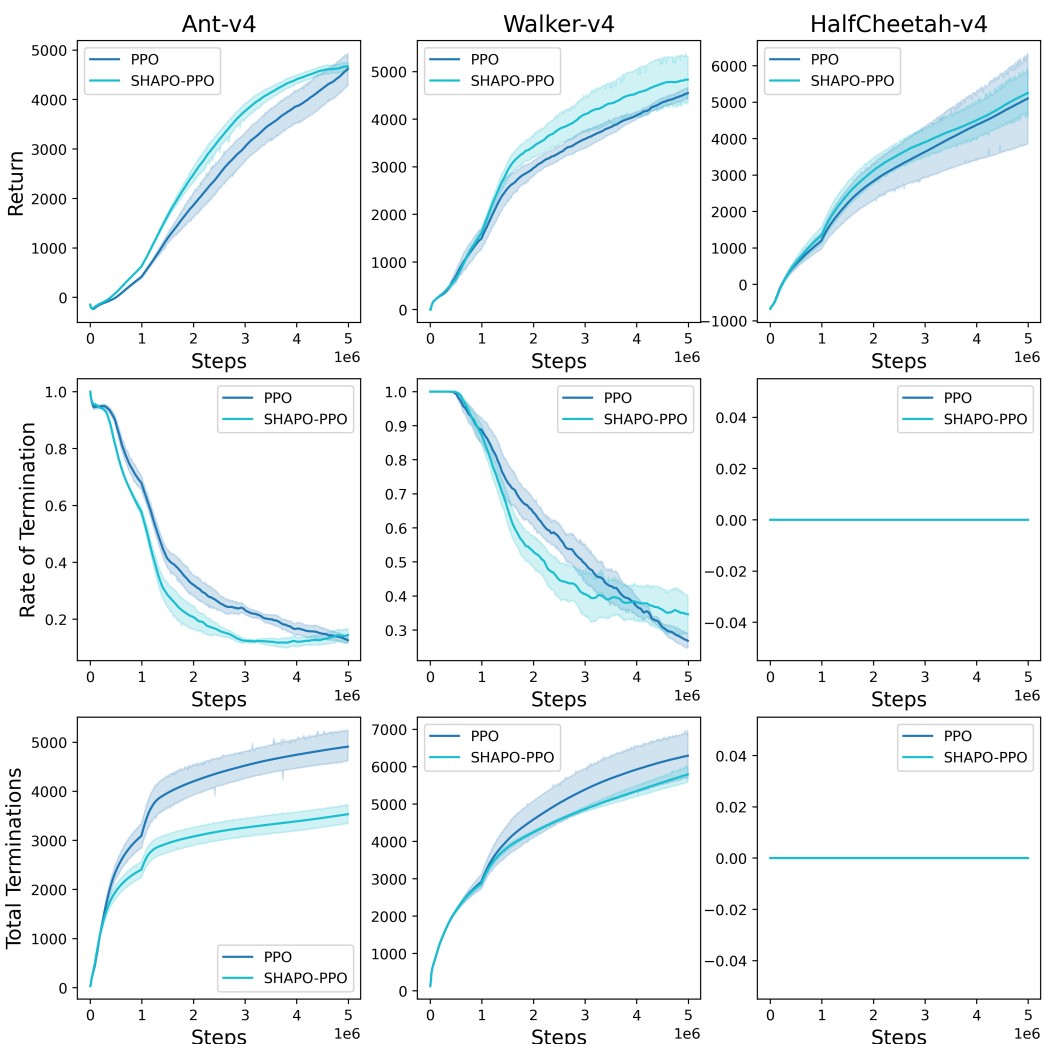

Figure 13: *SHAPO on Standard RL:* Columns correspond to *Ant-v4*, *Walker2d-v4*, and *HalfCheetah-v4*. Top row: episodic return; middle row: rate of termination; bottom row: total terminations. *SHAPO* yields substantial gains on *Ant-v4* and *Walker2d-v4*, where falling triggers early episode termination and creates a pronounced lower tail in the return distribution. In *HalfCheetah-v4*, which lacks termination or failure modes, the return distribution has no meaningful lower tail, and *SHAPO* consequently provides limited improvement.

Fig. 13 illustrates how the impact of *SHAPO* depends on the presence or absence of tail events in the environment. In *Ant* and *Walker2d*, *SHAPO* consistently improves the sample efficiency of the PPO baseline by reducing premature terminations; this allows agents to remain in the episode longer, resulting in fewer resets and more reward collected per rollout. In *HalfCheetah*, however, where no catastrophic failure or termination mode exists, the return distribution lacks a meaningful lower tail, and accordingly, *SHAPO* offers limited additional benefit. These results underscore that while *SHAPO* provides tangible advantages in settings with significant tail risk, its effect is naturally less pronounced in environments where undesirable events do not produce clear tail behavior in the return landscape.

## G    LLM USAGE

ChatGPT (Achiam et al., 2023) was used to polish writing and also as a tool to correct grammar and spelling mistakes.

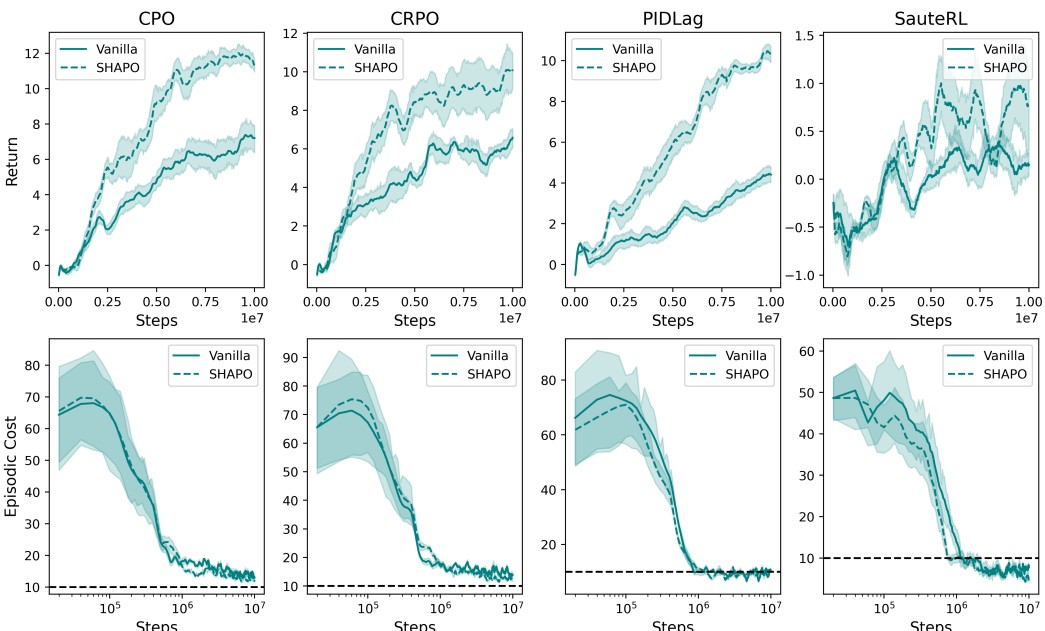

Figure 14: *PointGoal1*

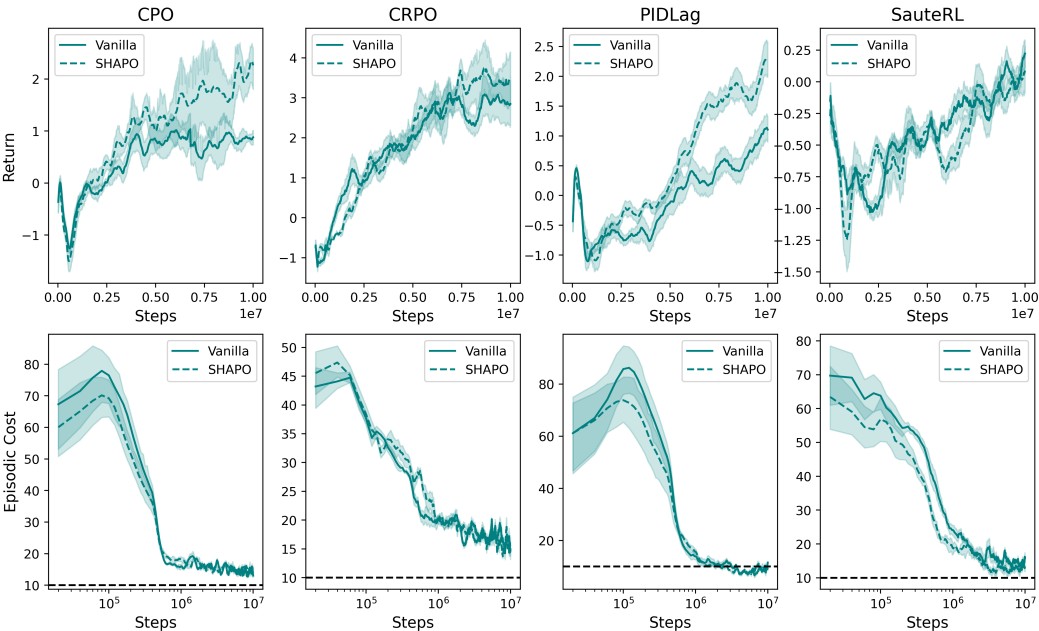

Figure 15: *PointButton1*

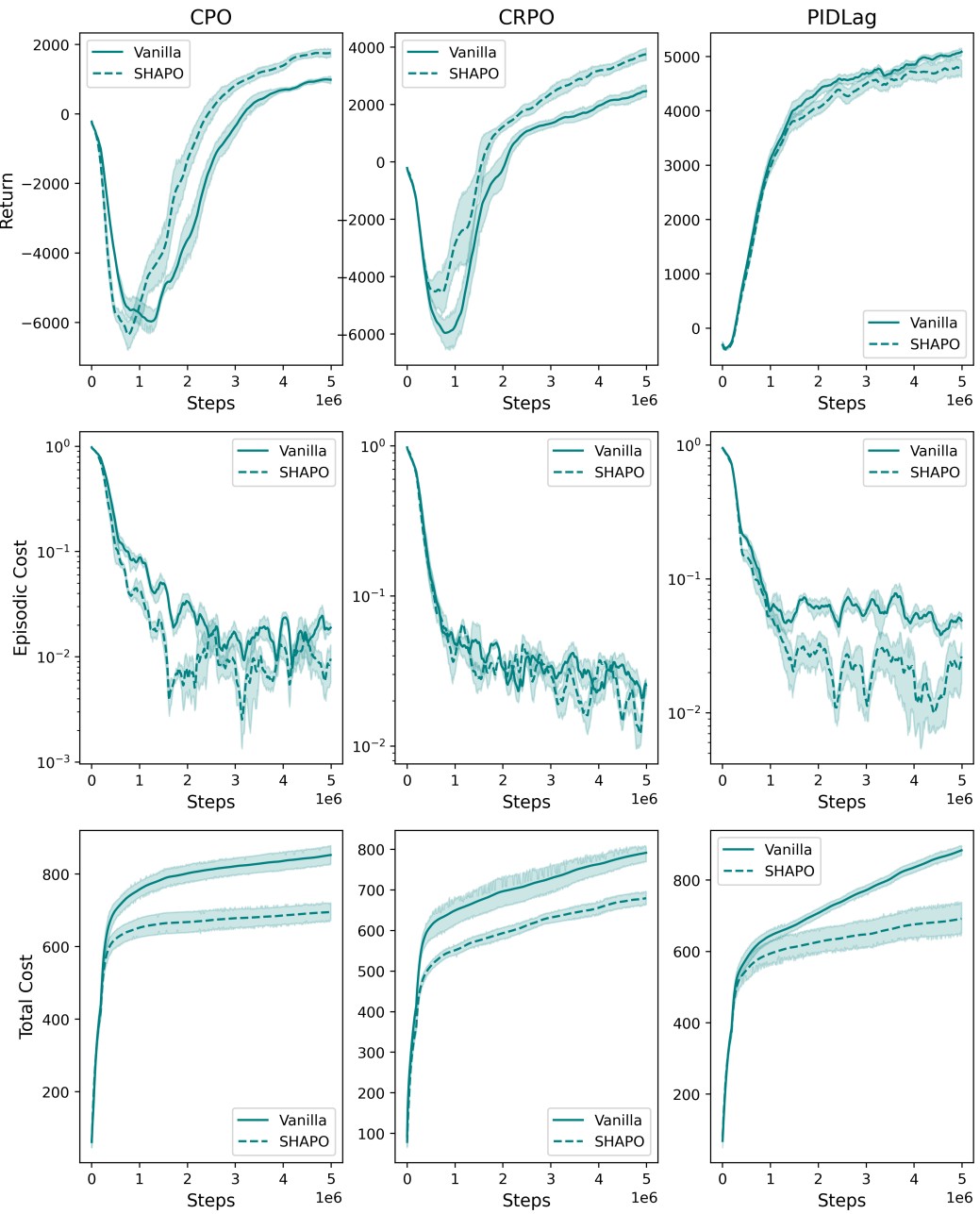

Figure 16: *Ant-v4*

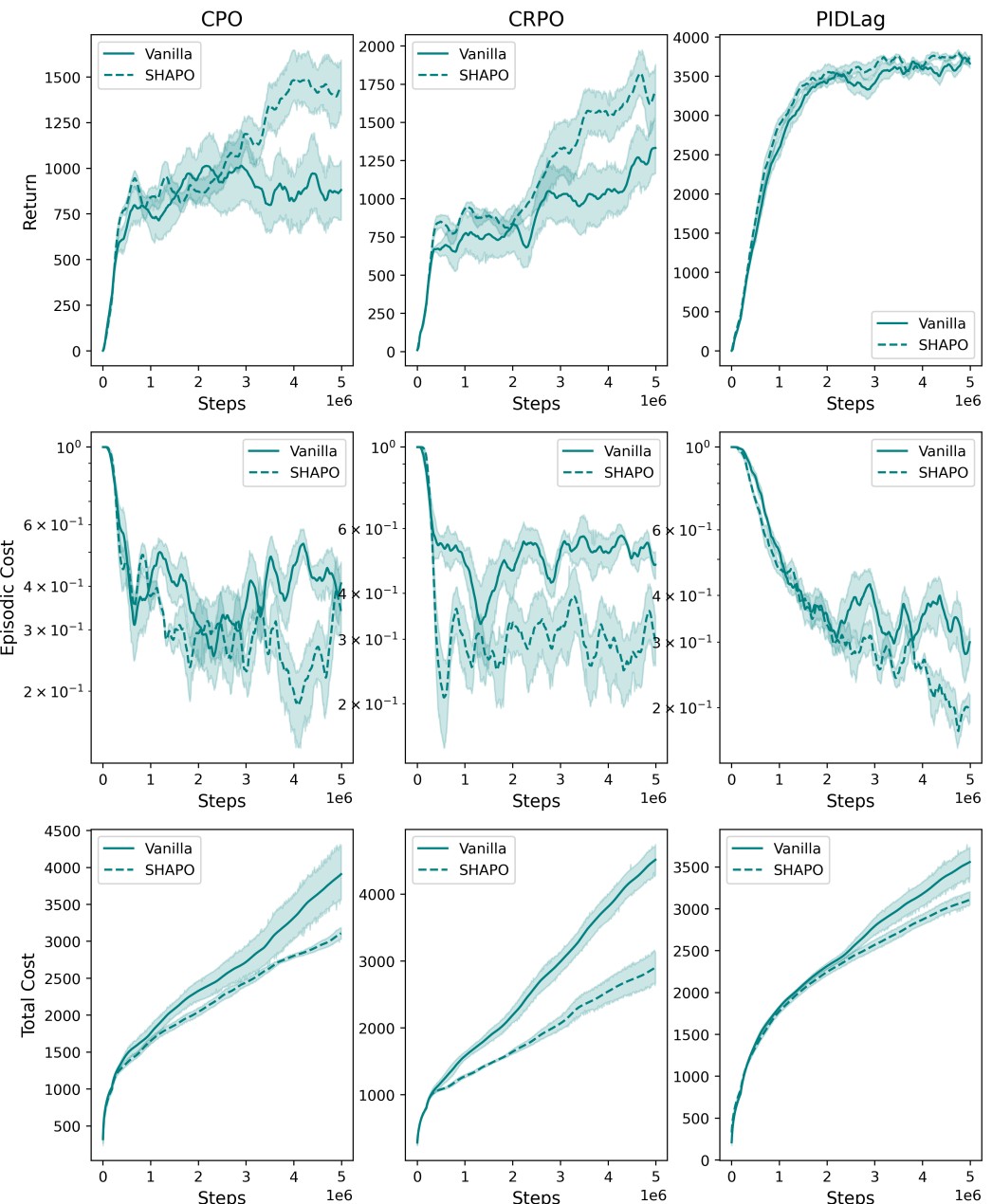

Figure 17: *Walker2d-v4*

