# OpenReview forum: "SHAPO: Sharpness-Aware Policy Optimization for Safe Exploration"
_ICLR.cc/2026/Conference — ICLR 2026 Poster_

### Official Review · Reviewer_huer · 2025-10-30

**Soundness:** 3
**Presentation:** 3
**Contribution:** 3
**Rating:** 6
**Confidence:** 3

**Summary:**

This paper introduces Sharpness-Aware Policy Optimization, a novel approach to enhance safe exploration in Reinforcement Learning by addressing the actor's epistemic uncertainty. The core idea is to use the "sharpness" (sensitivity of the policy to parameter perturbations) as a practical proxy for uncertainty, particularly in regions with scarce data.
SHAPO leverages principles from Sharpness-Aware Minimization (SAM). The proposed update rule operates in a pessimistic manner: it first finds the worst-case policy parameter θ within a local neighborhood that minimizes the performance objective (maximizes risk), and then computes the policy gradient at this "pessimistic" point. This results in a modification of the policy gradient that analytically promotes conservative behavior by amplifying the penalty for rare, unsafe actions while attenuating the reward for rare, safe actions. The method is implemented as a plugin on top of existing safe RL algorithms (TRPO) and demonstrates improved performance on the safety-efficiency Pareto frontier across various Safety Gym tasks.

**Strengths:**

1.  Applying Sharpness-Aware Minimization (SAM), typically used for improving generalization in supervised learning, to Safe RL is a  conceptually elegant contribution. This provides a new lens for addressing epistemic uncertainty in the actor network.
2. The analysis (Section 3.3) and empirical results support the claim that the SHAPO gradient effectively introduces a pessimistic bias. The method consistently achieves a lower frequency of catastrophic events and generally improves the safety performance of multiple SOTA safe RL baselines when used as a plugin.
3. The update rule is straightforward and can be readily integrated with different Policy Optimization algorithms without fundamental changes to their core optimization objective, showcasing high modularity.

**Weaknesses:**

1. The paper suffers from a crucial disconnect between the theory and implementation. Proposition 2 defines the perturbation magnitude \delta_{Down}​	 as a function of the sample size n and confidence level α. However, in the practical implementation and hyperparameter search (Appendix D), \delta_{Down}​ is treated as a fixed hyperparameter, which contradicts the theoretical guidance for an annealing schedule based on n.
2. The modeling of the posterior distribution Q(θ) is solely "motivated" by the BvM theorem. This represents a strong, unproven assumption, as the strict regularity conditions required for BvM are unlikely to hold in high-dimensional, non-convex deep RL settings, thus introducing a significant theoretical gap.
3. There is an explicit inconsistency in the definition of the covariance matrix for the distribution Q(θ) between the main text and the appendix proof of Proposition 5, which uses a distribution corresponding to a precision matrix proportional to $\sqrt n$. This error undermines the rigor of the derived relationship for $\delta _{Down}$.
​4. The paper focuses on comparing SHAPO's performance against standard constrained optimization methods (CPO, CRPO, etc.). To fully validate the claim of solving epistemic uncertainty in the actor, the baselines should ideally include methods that explicitly handle uncertainty or risk sensitivity, such as: Policy Gradient methods utilizing Dropout or Ensembles on the actor. Risk-Sensitive Policy Optimization methods (e.g., those based on CVaR or other risk measures).
5.  While $\delta _{Down}$ is the critical new hyperparameter of the method, the paper provides only a qualitative discussion of its sensitivity (Appendix D) and lacks a comprehensive, quantitative ablation study.

**Questions:**

1. Given the theoretical prescription that $\delta _{Down}$ should decay with the number of samples n, why was a fixed, tuned hyperparameter used in the implementation? Could the authors provide an ablation study comparing the fixed hyperparameter approach with the theoretically suggested annealing schedule?
2. Please clarify and correct the discrepancy in the definition of the posterior/likelihood distribution Q(θ) between Section 3.2 and Appendix B. A corrected, rigorous proof for Proposition 2 (or 5) is required.
3. To solidify the contribution of using "sharpness" as an uncertainty proxy, could the authors compare SHAPO's performance against policy gradient methods that explicitly model actor uncertainty or risk, such as an Ensemble Actor approach or a CVaR-based policy optimization method?

---

> ### Author Response · Authors · 2025-11-25
>
> ## [W1]: Theory Implementation Disconnect
>
> Please see the general response for all reviewers (Practical Implications of Proposition 2).
>
> ## [W2]: BvM theorem regularity conditions
>
> We agree with the reviewer that, for complex neural networks, the regularity conditions required for the classical Bernstein–von Mises (BvM) theorem are generally not satisfied.
> However, given that we lack a theory that model the complex learning process of the parameters of the actor in deep reinforcement learning, we find that the BvM framework still provides the best conceptual guide in this context.
>
>
> ## [W3 & Q3]: Definition of the covariance matrix for the distribution $Q(\theta)$ in appendix
>
> Thank you very much for your careful reading of the appendix and for pointing out this mistake. The definition of the covariance matrix of $Q(\theta)$ in the appendix has been corrected and now matches the one given in the main paper. We have also corrected the proof of Proposition 5 in the appendix accordingly. This error did not affect the derivation of the relationship $\delta = \frac{z^2_\alpha}{2n}$ which is stated in the main paper in Proposition 2.
>
> We agree that we should ideally compare against methods that explicitly model the parameter uncertainty of the actor. However, to our knowledge, existing work on uncertainty in safe RL enforces pessimism in the actor by estimating uncertainty over the value function. Such methods like WCPG[1] and Ensemble Critic do not model the uncertainty of the actor and therefore differ fundamentally from our approach that focuses on the actor.  Nonetheless, we have implemented on-policy version of WCPG and Ensemble Critic and we provide results for these approaches in Appendix F. Since our method only intervenes on the actor's update, it is compatible with these two approaches that focus on the critic. We therefore combine our method with these approaches and show that this consistently leads to improvement. Moreover SHAPO alone leads to best performance, even though more effort could certainly be done to get the best of these approaches on these environments.
>
> [1] Tang, Yichuan Charlie, Jian Zhang, and Ruslan Salakhutdinov. "Worst Cases Policy Gradients." Conference on Robot Learning. PMLR, 2020.
>
> ## [W4]: Hyperparameter Sensitivity
>
> Please refer to the general response to all reviewers.
>
> ## [Q1]: $\delta_{Down}$ decay
>
> We have provided an ablation study in Section 6.1, with a detailed discussion and analysis in Appendix E of the revised manuscript. In these experiments, we assume that the effective sample size $n$ corresponds to the total number of state-action pairs experienced by the agent. We compare two settings: using a fixed $\delta_{down}$ versus a decaying $\delta_{down}$ proportional to $1/(2n)$ which corresponds to keeping $z_{\alpha}$ fixed or $\alpha$ fixed, respectively.
>
>
> ## [Q2]: Discrepancy in definition of $Q(\theta)$
>
> Thanks for pointing it out. We’ve corrected it in the revised manuscript.

---

### Official Review · Reviewer_tvhm · 2025-11-01

**Soundness:** 3
**Presentation:** 3
**Contribution:** 3
**Rating:** 8
**Confidence:** 4

**Summary:**

The paper explores the use of sharpness aware optimization for safe RL. Sharpness aware optimization introduces a max-min problem instead of maximization to flatten the optimization profile. In my understanding this max-min problem would aim to prefer a flatter maximum to a sharp peak maximum in the parameter space. This approach aims to improve robustness of the optimization solution to epistemic uncertainty. The authors conduct a series of experiments that show an improvement of performance of different algorithms with and without SHAPO.

**Strengths:**

1.The idea is novel to my best knowledge and quite interesting.
2. The paper contains a deep discussion on my sharpness aware optimization is a good fit for ML problems in general. In section 3.3, the authors dive deeply into why SHAPO can be a good fit for RL.
3. The experiments and ablation study are great

**Weaknesses:**

I couldn't come up with many weaknesses, but here are a couple
* The policy update section 3.1 is building the solution bottom-up, but maybe a top-down approach would be slightly easier to read.
* It would be good to explain how the policy update is actually implemented. The authors present a quadratic optimization problem in  TRPO style update and state that SHAPO update can be applied to any on-policy algorithm. Can the authors elaborate how SHAPO can be applied to a PPO update?

**Questions:**

* The approach makes total sense and it seems relevant not only for safe RL, but also for RL. What happens if SHAPO is applied without a safety constraint? Did the authors perform this ablation study?
* Can the authors provide similar Figures to Fig 6 and 7, but in terms of episode return vs episode cost? I think their decision to focus on cost rate is correct, but having a full picture (maybe in appendix) would be good.
* Can the authors elaborate what’s the base method for Saute RL? As the authors are aware, Saute RL can be applied to any RL algorithm.
* Safety gymnasium has many more environments that could be useful for your future research and evaluations. Safety Gymnasium: A Unified Safe Reinforcement Learning Benchmark https://arxiv.org/abs/2310.12567
* What’s the additional computational burden that the approach adds to the algorithm?

---

> ### Author Response · Authors · 2025-11-25
>
> Thank you for your valuable and very positive feedback.
>
> ## [W1]: Bottom-up vs Top-Down and SHAPO for on-policy RL algorithms
>
> We agree with you that adding a more top–down presentation of the method could improve readability. In this direction we have now added Subsection 4.4 in the revised manuscript that provides more information about how our method applies to any on-policy RL algorithm, abstracting the modification in Algorithm 1 and providing pseudo-code in Appendix A on how to modify each baseline algorithm. We hope these changes makes our method more directly accessible to readers.
>
>
> ## [W2]: Applying SHAPO to PPO
>
> PPO can be viewed as a first-order approximation to TRPO, replacing the explicit KL-divergence constraint with a clipped likelihood-ratio objective that implicitly restricts the updated policy to remain within an $\epsilon$-trust region of the old policy.  A straightforward way to apply SHAPO to the PPO algorithm consists of using Algorithm 1 to get the gradient of the PPO clipped objective at the perturbed parameter. We are presenting pseudo-code for SHAPO-PPO in Appendix A (Algorithm 4) of the revised manuscript.
>
>
> ## [Q1]: SHAPO on unconstrained RL
>
> You are correct that SHAPO can in principle be applied to standard (unconstrained) RL settings as well. In such cases, the lower tail of the return or loss distribution naturally corresponds to undesirable or ``unsafe'' outcomes, which are typically penalized through negative rewards. In fact, Lagrangian TRPO can be viewed as unconstrained RL with an augmented reward function in which cost-inducing states receive additional negative reward proportional to the Lagrange multiplier $\lambda$. From this perspective, the SHAPO mechanism remains applicable, as it directly regulates sensitivity to tail events in the actor's loss landscape.
>
> While our work focuses on safe exploration and the constrained RL setting, we agree that extending SHAPO to unconstrained RL is natural and potentially beneficial. Due to time constraints, we have not included experiments in that regime, but we believe the approach should carry over straightforwardly.
>
>
> ## [Q2]: Episodic Return vs Episodic Cost plots
>
> Thank you for the suggestion. We would like to note that Figures 14--17 in the appendix already include plots of episodic return, episodic cost, and total cost across training steps.
>
> ## [Q3]: Base Method for Saute RL
>
> You are right in stating that Saute RL can be applied to any RL algorithm. We indeed didn’t specify this detail in the original paper. In our experiments, Saute RL is using TRPO as the underlying RL algorithm. We’ve added this detail in the revised paper in Section 5 (Baselines).
>
> ## [Q4]: More environments in Safety Gymnasium
>
> We agree that we can expand our experiments to more environments in the future. In the interest of time, our focus in the paper has been to demonstrate its versatility on locomotion (Mujoco environments) and navigation (PointGoal1, PointButton1) environments as well as providing adequate analysis to understand the details of the method.
>
> ## [Q5]: Computational Burden of SHAPO
>
> Please see the general response to all reviewers (Runtime Analysis).

---

> > ### Comment · Reviewer_tvhm · 2025-11-26
> >
> > Thank you for your responses and clarification!
> >
> > I think having an experiment with PPO would increase applicability of this approach.
> >
> > I was hoping that there's a specific reason that SHAPO is not used in the unconstrained setting. For example, hypothetically, the learned shapo return is not higher than vanilla return.
> >
> > Having these experiments would make the paper even stronger, but I think the current results are publishable as they are.

---

> ### Author Response · Authors · 2025-11-28
>
> Thank you for your comment. We agree that including PPO experiments in unconstrained RL setting strengthens the applicability of our approach. To address this, we have added results for PPO and _SHAPO_-PPO on three MuJoCo environments in standard (unconstrained) RL setting: _Ant-v4_, _Walker2d-v4_, and _HalfCheetah-v4_ in **Appendix G and Figure 14** of the revised manuscript. _SHAPO_ provides substantial gains on _Ant-v4_ and _Walker2d-v4_, where falling triggers early termination and creates clear tail events. In _HalfCheetah-v4_, however, there is no termination or failure mode, so the return distribution lacks a pronounced lower tail; accordingly, _SHAPO_ offers only limited improvement.

---

### Official Review · Reviewer_oXHA · 2025-11-06

**Soundness:** 3
**Presentation:** 3
**Contribution:** 2
**Rating:** 6
**Confidence:** 3

**Summary:**

The paper studies the application of sharpness awareness minimization (SAM) in policy optimization. SAM aims at optimizing the model such that the worst-case loss in its neighboring parameters is low, encouraging the optimizers to find low-loss solutions that are also flat. In particular, the authors extend Fisher-SAM, a prior SAM method that leverages Fisher information matrix to estimate the local geometry to efficiently estimate the local loss-maximizing parameter, to trust-region policy optimization for safe exploration. The paper provides an interpretation of Fisher-SAM as a pessimistic estimation of the loss subject to the uncertainty quantified by the Fisher matrix. This in turn allows the authors to interpret the worse-case expected return in the neighboring parameters informed by the Fisher matrix as an uncertainty lower-bound on the actual expected return. The paper evaluates the proposed method, SHAPO, on safety gym and shows that the proposed method outperforms prior methods in terms of the return/cost trade-off.

**Strengths:**

- The paper is easy to read and the proposed method is well-motivated and technically sound.
- The empirical results show statistically significant improvements of the proposed method over prior safe RL methods. Ablations are thorough and show that major components of the algorithms all contribute to the effectiveness of the proposed method.
- Even though SAM/Fisher-SAM is not new, the application of it in the context of safe RL is new and seems effective.

**Weaknesses:**

- The proposed method requires solving the natural gradient direction that maybe very expensive.
   - I tried to look for the implementation details in the paper but could not find it. It would be good if the authors could include more implementation details in the paper for better reproducibility.
   - I checked the linked anonymous codebase briefly and it seems that the implementation involves running an iterative conjugate gradient procedure (https://anonymous.4open.science/r/Safe-Policy-Optimization-813E/safepo/single_agent/shapo.py). This procedure can incur a non-trivial amount of computation overhead. It would be good if the authors could include more details on the run-time of the algorithm compared to prior methods in the paper.

- Hyperparameter sensitivity analysis is missing from the empirical evaluations. In Appendix, the authors mentioned that "Most common choice of SHAPO hyperparameters that gave the best performance across tasks and baselines was $\delta_{\mathrm{Down}} = 0.0001$ and $\rho_{\mathrm{critic}} = 0.01$", but it is unclear how different hyperparameters influence the performance of the algorithm. Including some analysis on the sensitivity of the performance with respect to these hyperparameters (especially $\delta_{\mathrm{Down}}$ could help gain a better understanding of the robustness and effectiveness of the algorithm.

**Questions:**

- In Figure 3, 6 and 7, how are the shapes of the circles around the mean/average performance determined?
- In algorithm 2, how is $U_\theta$ computed?

---

> ### Author Response · Authors · 2025-11-25
>
> We thank you for your constructive and valuable comments on our work.
>
> ## [W1]: Implementation Details
>
> We acknowledge that the original paper lacked implementation details in some areas. We’ve provided more details on how our method applies on every on-policy RL algorithms in Subsection 4.4 of the revised manuscript and pseudo-code for SHAPO versions of the baseline algorithms used in Appendix A. Also our code is publicly available and we are ready to provide more details that you may find necessary.
>
> ## [W2]: Runtime Analysis
>
> Please see the general response to all reviewers.
>
>
> ## [W3]: Hyperparameter Sensitivity Analysis
>
> Please see the general response to all reviewers.
>
> ## [Q1]: Empirical Results
>
> Thank you for catching this, width and height of the ellipses represent the standard error over $5$ seeds. This has been added in the caption of Figure 2, as well as in Section 5 (Metrics).
>
> ## [Q2]: How is $U_{\theta}$ computed in Algorithm 2 (now Algorithm 3 in the revised paper)?
>
> In our revised manuscript, we now write $U_F$ (instead of $U_\theta$) to emphasize the dependency on the Fisher matrix. The analytical solution to this optimization problem is provided and proved in Lemma 4 in Appendix B. In practice, we rely on TRPO’s implementation as provided in the code base [Safe-Policy-Optimization](https://github.com/PKU-Alignment/Safe-Policy-Optimization), which involves computing the Fisher vector product to avoid computing directly the inverse of the Fisher matrix.

---

> > ### Comment · Reviewer_oXHA · 2025-11-25
> >
> > Thanks for your response! All my concerns have been addressed and I have raised my score to 8.

---

### Official Review · Reviewer_2kQM · 2025-11-07

**Soundness:** 3
**Presentation:** 3
**Contribution:** 3
**Rating:** 6
**Confidence:** 3

**Summary:**

This paper introduces Sharpness-Aware Policy Optimization (SHAPO) to address the challenge of safe exploration in safety-critical RL. The core idea is to use the actor's sharpness as a practical proxy for epistemic uncertainty. The method implements a pessimistic policy update. In each step, it first solves an inner-loop optimization to find the "worst-case" perturbed parameters ($\theta_0 + \epsilon_{Down}$) that minimize the policy objective within a trust region which is defined by the Fisher Information Matrix (KL divergence). It then computes the policy gradient ($\tilde{g}$) at this worst-case point and uses this pessimistic gradient for the final policy update .
Analytically, the authors show this process implicitly reweighs gradients, amplifying the effect of rare, unsafe actions and tempering the effect of rare, safe actions.

**Strengths:**

- The paper tackles the critical challenge of safe exploration in a practical and useful manner, which is a prerequisite for real-world RL deployment.

- The analytical insights, including Proposition 3 on gradient adjustments for rare actions and the reinterpretation of perturbations as pessimistic quantiles (Proposition 2), are clear and well-supported.

- Evaluations across multiple baselines and environments demonstrate consistent gains, with Pareto improvements and reduced catastrophic failures

**Weaknesses:**

-  Major Theory-Implementation Mismatch

The paper's theoretical justification for being "uncertainty-aware" (presented in L249-256)  hinges on the idea that the inner trust region $\delta_{Down}$ should adapt based on the amount of data $n$. However, the actual implementation described in Appendix D uses a fixed, grid-searched $\delta_{Down}$.

- Gap in Intuitive Justification

The paper argues its strength comes from using the Fisher metric . However, the core intuition for why it is safe (Section 3.3)  is derived using a simplified Euclidean metric (Appendix C, L778: "For simplicity, we consider here a Euclidean perturbation...").

**Questions:**

- Could you clarify whether the adaptive scaling rule was implemented or tested in any form?
If not, how do you reconcile this discrepancy between the theoretical motivation and the actual implementation?
- Could you prove that the core safety intuition from Figure 2  holds for the Fisher-based update, not just the simplified Euclidean case  shown in the appendix?

---

> ### Author Response · Authors · 2025-11-25
>
> We thank you for providing a very constructive and thorough review of the paper.
>
> ## [W1 & Q1]: Theory-Implementation Mismatch
>
> Please see the general response for all reviewers (Practical Implications of Proposition 2).
>
> ## [W2 & Q2]: Gap in Intuitive Justification
>
> Thank you for pointing this out. The analysis provided in the paper is for a simple Gaussian policy with a fixed variance of $1$. For this parametrization based on the mean only, the Kullback-Leilbler divergence is equal to half the square of the Euclidean distance between the means: $\text{KL}(\pi(\cdot;\mu_0,1)\Vert \pi(\cdot;\mu_0+\epsilon_\text{Down},1))=\epsilon_\text{Down}^2$. Therefore, the approach based on the Kullback-Leibler neighborhood ($\text{KL}(\pi(\cdot;\mu_0,1)\Vert \pi(\cdot;\mu_0+\epsilon_\text{Down},1))\leq \epsilon_\text{Down}$) coincides with the one using the Euclidean metric on parameters ($\epsilon_\text{Down}\leq \rho$) and both can be expressed in terms of $\rho=\sqrt{2\delta_\text{Down}}$. Proposition 3 therefore states that applying Sharpness Aware Optimization (with Euclidean or KL bounds) to the actor results in a re-weighting of gradients which emphasizes rare unsafe transitions over safe ones.
> We’ve corrected this in the revised manuscript (see Appendix C and around Proposition 3).

---

### Author Response · Authors · 2025-11-25
**General response to all reviewers**

We thank all the reviewers for providing a very constructive and thorough review of the paper.
It has been very helpful in improving the quality and clarity of the paper. We have now revised the manuscript and all the modifications to the original manuscript have been highlighted in teal for ease of reference. Below we address three main concerns that have been raised by at least two reviewers. Other specific questions or concerns have been addressed individually.

## Practical Implications of Proposition 2

We recognize that the original version of our manuscript was not clear enough about the practical implications of Proposition 2. We want to stress that the main implication of Proposition 2 is to ensure that our method effectively enforces some level of pessimism in face of epistemic uncertainty about the actor, for any positive value of $\delta_\text{Down}$. We have attempted to clarify this important point through new paragraphs or sentences in Subsections 4.2, 4.4 and 6.1 of the revised paper. Crucially, we consider that while Proposition 2 allows for a very valuable interpretation of the value of $\delta_\text{Down}$ used at each update step of our algorithm in terms of a confidence level $\alpha$ and an effective sample size $n$, choosing the appropriate level of pessimism for each update step during training is not straightforward. The reason for this is two-fold:

- it is challenging to estimate the effective sample size $n$ in continuous-control reinforcement learning where data are temporally correlated and only a subset of collected samples provide independent information.
- Given the complex coupling between the actor and critic, together with the challenges posed by exploration, it is difficult to determine the appropriate level of pessimism (i.e. the value of $\alpha$) to adopt at each stage of the training for the task of safe exploration.

In this context, considering the value of $\delta_{Down}$ as tunable hyperparameter and applying this value constantly during training offer a simple yet effective strategy, which we decided to adopt.

However, we agree that investigating other schemes and schedules for the level of pessimism to adopt during the training of our actor is an important and interesting avenue. We therefore added some analysis and results in this sense in Subsection 6.1 and Appendix E of the revised manuscript. To leverage Proposition 2 and evaluate the level of pessimism corresponding to a value of $\delta_\text{Down}$, we consider for each update step $t$ the number $n_t$ of state-action pairs collected so far by our agent as a proxy for the effective sample size $n$.  Our simple strategy using a constant $\delta_\text{Down}$ can therefore be interpreted as increasing the level of pessimism as the agent learns. Indeed, the percentile $z_\alpha^t=-\sqrt{2n_t\delta_\text{Down}}$  corresponds to a decreasing level of confidence $\alpha_t$ as $n_t$ increases. This means that as the agent gathers more information about the world, we increasingly adopt a more pessimistic view of the epistemic uncertainty associated with it. This makes sense for practical reinforcement learning where excessive pessimism in the early phase of RL training can lead to conservative behaviours, while pessimism at the later stages can ensure better constraint satisfaction and lower total cost as a result.

In order to explore a different pessimism schedule, we investigated an adaptive scaling rule where a pessimism value $\alpha\in(0,0.5)$ is selected as a hyperparameter and the bound $\delta_\text{Down}=\frac{z_\alpha^2}{2n_t}$ is computed according to Proposition 2. This approach effectively relies on a bound that decreases as the agent learns. Results (presented in Figure 8) show that both strategies (constant $\delta_{Down}$ and based on $\alpha$) lead to improved performance over the vanilla baseline.

## Hyperparameter Sensitivity Analysis

We agree that hyperparameter sensitivity analysis can indeed provide more insights into the practicality and the robustness of SHAPO. We’ve provided a detailed hyperparameter analysis in Appendix D.2 and Figure 9 in the revised manuscript. The results show that very low values of $\delta_{Down}$ are equivalent to the vanilla algorithm, very high values of $\delta_{Down}$ leads to unstable learning and poor performance, and that there are several intermediate values which lead to consistent performance improvement.

## Runtime Analysis

We acknowledge that the method results in some additional computational overhead in terms of an additional inner TRPO step (which involves conjugate gradient computation). From our experiments we’ve observed that this overhead is insignificant given the extensive amount of compute time dedicated to data collection.  We’ve provided the results for the overall runtime of the algorithm as well as time taken for a single update step in Appendix D.3 and Table 1 of the revised manuscript.

---

### Author Response · Authors · 2025-12-04

We believe we have comprehensively addressed the reviewers' concerns and suggestions in the revised manuscript. Specifically, we bridge the perceived gap between the theory and practical implementation by providing a more detailed discussion and additional experiments in **Sections 4.2, 4.4, 6.1 and Appendix E**.

Furthermore, we strengthened the empirical evaluation by adding hyperparameter sensitivity and runtime analyses, while also demonstrating the versatility of our method on unconstrained RL algorithms in **Appendices D-G**. We thank the reviewers for their insightful feedback, which we believe has significantly improved the quality of the paper.

---

### Meta-Review · Area_Chair_wY5N · 2025-12-27

**Summary:**

The reviewers initially recognized the novelty of applying Sharpness-Aware Minimization (SAM) to the actor's update for safe exploration but raised several significant concerns regarding the gap between theory and practice. Specifically, reviewers (2kQM, huer) questioned the disconnect between the theoretical motivation—which suggests an adaptive penalty based on sample size (via the Bernstein-von Mises theorem)—and the practical implementation using a fixed hyperparameter. Additionally, reviewers requested more comprehensive empirical evidence, including sensitivity analyses (OXHA), runtime analysis (OXHA), comparisons against explicit uncertainty-aware baselines like WCPG (huer), and demonstrations of applicability to unconstrained settings like PPO (tvhm).
The authors effectively addressed these concerns in the rebuttal by providing comprehensive sensitivity and runtime analyses, clarifying theoretical inconsistencies, and expanding experiments to include uncertainty-aware baselines and unconstrained settings. These additions significantly strengthened the empirical rigor and better aligned the practical implementation with the theoretical motivation. Given the reviewers’ consensus on the method’s value for safe exploration, we recommend acceptance.

**Reviewer Concerns:**

Most of the major reviewer concerns were addressed in the rebuttal. In response to Reviewer OXHA, the authors added a thorough sensitivity analysis for $$\delta_{down}$$ and $\rho_{critic}$, along with a runtime study showing that the additional computational cost is negligible. Reviewer huer’s request for stronger baselines was met by including comparisons with explicit risk-sensitive methods such as Worst-Case Policy Gradients and Ensemble Critics. The authors also followed Reviewer tvhm’s suggestion to evaluate unconstrained settings, adding PPO experiments on Ant, Walker2d, and HalfCheetah, which demonstrate consistent reductions in tail risk even without safety constraints. On the theory side, the covariance matrix inconsistency noted by Reviewer huer was corrected, and the intuition gap raised by Reviewer 2kQM regarding Fisher versus Euclidean metrics was clarified by explaining their equivalence under specific parameterizations. Overall, no major concerns remain outstanding.

**Reviewer Scores:**

Reviewer OXHA: 6 → 8. This reviewer explicitly requested runtime and sensitivity analyses, and stated that these additions resolved their main concerns. Given that the rebuttal directly addressed these points in detail, it is reasonable to believe their score would have increased to a clear accept.
 Reviewer tvhm: 8 → 8. This reviewer was already strongly positive. While the additional PPO and unconstrained experiments further strengthened the paper, they are unlikely to change an already high score.
 Reviewer huer: 6 → 6. Although the authors corrected the theoretical inconsistency and added the requested uncertainty-aware baselines, this reviewer emphasized rigor on both theory and uncertainty modeling. It is conservative to assume that, while concerns were mitigated, the score may have remained unchanged.
 Reviewer 2kQM: 6 → 6. The rebuttal clarified the motivation behind the theory–practice connection, but given the reviewer’s original skepticism about the theoretical grounding, it is reasonable to assume the clarification alone may not have been sufficient to justify a score increase.

---

### Decision · Program_Chairs · 2026-01-26

Accept (Poster)